# Earthquake Contributions to Coastal Cliff Retreat

Colin K. Bloom[1], Corinne Singeisen[1], Timothy Stahl[1], Andrew Howell[1,2], Chris Massey[2]

[1]School of Earth and Environment, University of Canterbury, Christchurch, 8041, New Zealand
[2]GNS Science, Avalon, Lower Hutt, 5010, New Zealand

*Correspondence to*: Colin K. Bloom (colin.bloom@pg.canterbury.ac.nz)

**Abstract.** Modeling suggests that steep coastal regions will experience increasingly rapid erosion related to climate change induced sea level rise. Earthquakes can also cause intense episodes of coastal cliff retreat, but coseismic failures are rarely captured in the historical record used to calibrate most cliff retreat forecast models. Here, we disaggregate cliff-top retreat related to strong ground motion and non-

seismic sources, providing a unique window into earthquake contributions to multidecadal coastal cliff retreat. Widespread landsliding and up to c. 19 m of coastal cliff-top retreat occurred in the area of Conway Flat during the 2016 Kaikōura (New Zealand) earthquake despite relatively low (c. 0.2 g) peak ground accelerations. While coastal cliff-top retreat has been spatially and temporally variable over the historical record, aerial imagery suggests that large earthquake induced landslide triggering events

disproportionately contribute to an average 0.25 m/year retreat at Conway Flat. The 2016 Kaikōura earthquake represents c. 24% of the total cliff-top retreat over 72 years and c. 39% of cliff-top retreat over 56 years. Additionally, we infer that significant retreat between 1950 and 1966 is the result of local seismicity. Together these two events account for c. 57% of cliff-top retreat over 72 years. Earthquake-related debris piles at the base of the cliffs have been rapidly eroded since the 2016 Kaikōura

earthquake (more than 25% loss of debris volume in 5 years) and there will likely be little evidence of the earthquake within the next decade. In regions with similar lithologic and coastal conditions, evidence of past widespread single-event cliff-top retreat may be limited or non-existent. The results demonstrate that cliff-top retreat projections using historical records may significantly underestimate true retreat rates in seismically active regions.

## 1 Introduction

Regional coastal modeling suggests an increasing rate of coastal cliff retreat as sea level rises from climate change (e.g. FitzGerald et al., 2008; Limber et al., 2018). This increasing retreat rate will pose a significant hazard to people and property around the globe, particularly in regions that face a high risk due to population exposure (e.g. He and Beighley 2008). The response of individual coastal cliffs to sea

level rise is complicated by a range of feedbacks and site-specific conditions, for example changing beach volume, the transport of failed material from more erosive sections of coastline, and cliff material strength (e.g., Dickson et al., 2007; Ashton et al., 2011) as well as by the temporal variability of cliff retreat (e.g., Hall et al., 2002; Hapke and Plant 2010). Many decadal to multidecadal models of coastal

cliff retreat rely heavily on historical records and legacy aerial imagery (typically less than 50-100

years) for calibration, in part to capture some of this spatial and temporal variability (e.g. Dickson et al.,

2007; Young et al., 2014; Limber et al., 2018). Unfortunately, direct evidence of past coastal failures is

rarely preserved in the active coastal environment (Francioni et al., 2018) making it difficult to confirm

that the historical record is representative of all possible preconditioning and triggering mechanisms for

coastal cliff collapse. This is particularly important when considering that the cliff face may erode at

different relative rates over decadal to multidecadal timescales. For example, rainfall-induced landslides

may erode the cliff-top faster than coastal erosion from wave action at the base of the cliff, the latter of

which ultimately dictates the longer-term pace and spatial pattern of cliff instability.

In tectonically active regions, earthquakes can cause widespread coastal cliff collapse (Griggs and

Plant, 1998; Hancox et al., 2002) but their contribution to coastal cliff retreat has yet to be considered in

most decadal to multidecadal forecasts (Hapke and Richmond, 2002). Beyond the logistical challenge

of incorporating infrequent and spatially variable strong ground motion as an input in coastal cliff

retreat models, the extent to which earthquakes influence multidecadal coastal cliff retreat remains

unclear at most sites.


The 2016 $M_w$ 7.8 Kaikōura earthquake on the South Island of New Zealand (Figure 1) triggered

hundreds of landslides along coastal slopes, including areas of coastal cliff-top collapse under relatively

low ground motion conditions (< 0.2 g PGA) (Massey et al., 2018). We use pre- and post-event aerial

imagery at Conway Flat, an c. 8 km stretch of the Kaikōura coast where widespread failure from the

2016 Kaikōura earthquake was observed in coastal cliffs, to quantify the influence of earthquake related

cliff-top retreat and disaggregate strong ground motion related retreat from non-seismic related retreat.

Additionally, the volume of failed debris removed from the beach at Conway Flat by coastal erosion

following the 2016 Kaikōura earthquake is calculated to demonstrate how quickly evidence of a large

single-event cliff retreat is lost in the active coastal setting. While the conditions described here may not

apply to all coastal cliffs, the results provide a template for further investigation of coastal cliffs in

tectonically active regions.

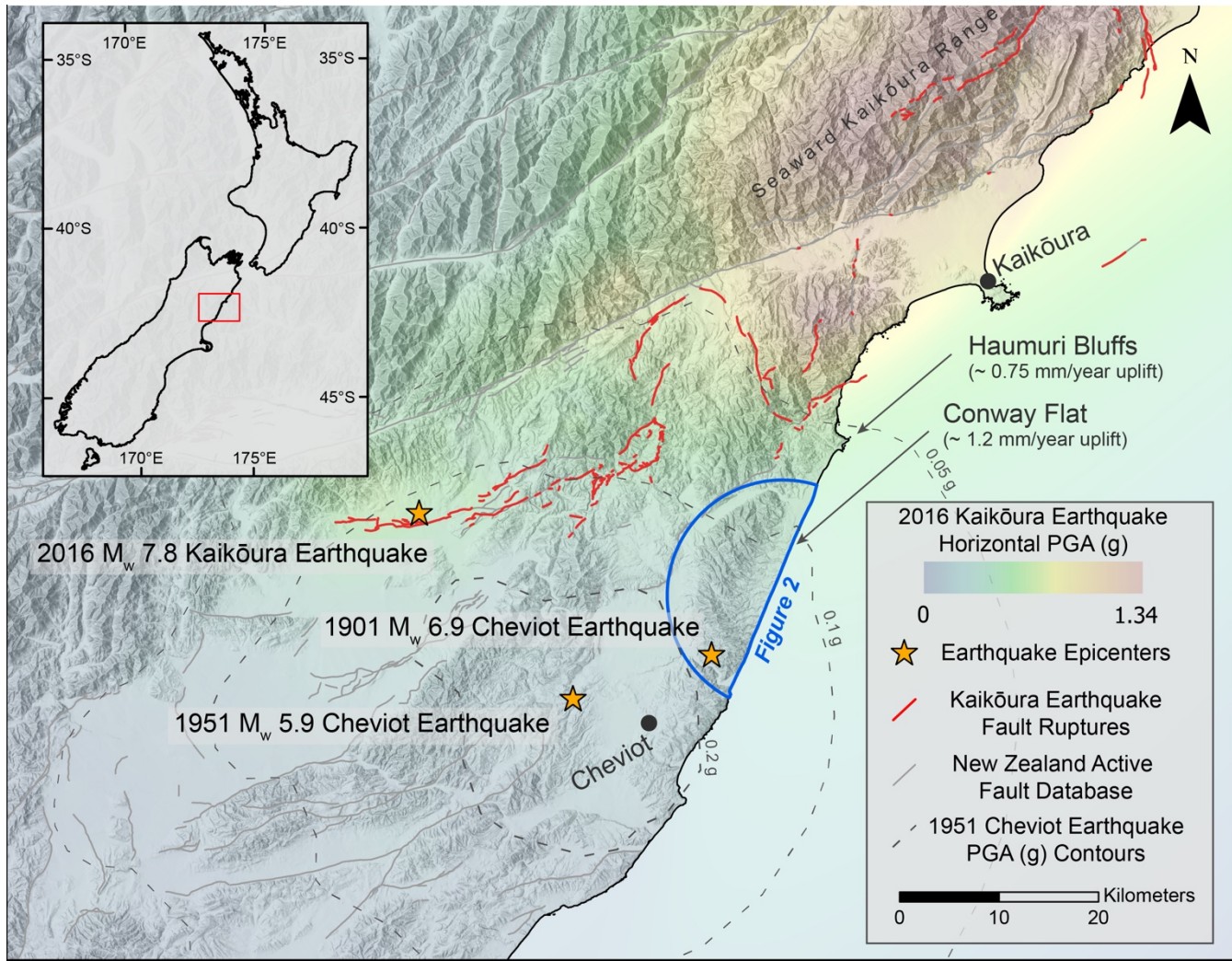

**Figure 1: Overview of the Kaikōura Coast.** Horizontal Peak Ground Acceleration (PGA) from the 2016 Mw 7.8 Kaikōura Earthquake (Bradley et al., 2017) is shown as a color ramp over a multidirectional hillshade derived from an 8 m DEM (LINZ, 2022). PGA from the 1951 Mw 5.9 Cheviot Earthquake (ShakeMapNZ; Horspool et al., 2015) is shown as dashed grey contours radiating away from the earthquake epicenter. Faults in the New Zealand Active Fault Database (Langridge et al., 2016) are shown as solid grey lines while faults that ruptured to the surface during the 2016 Kaikōura earthquake (Litchfield et al, 2018) are shown as solid red lines. Late Pleistocene uplift rates are reported at Haumuri Bluffs and Conway Flat (Barrell et al., 2022). The location of Figure 2 is shown as a solid semi-circular blue outline.

## 2 Background

### 2.1 2016 Mw 7.8 Kaikōura Earthquake

The 2016 Mw 7.8 Kaikōura earthquake initiated on the Humps fault c. 40 km inland from the coast in the northeastern South Island of New Zealand. Over approximately two minutes, fault rupture propagated onto more than 20 on- and off-shore faults primarily to the northeast of the epicenter (Figure 1; Litchfield et al., 2018). The earthquake generated more than 30,000 landslides which were primarily concentrated within the steep slopes of the Seaward Kaikōura Range, around surface fault ruptures, and in steep sections of coastline including the coastal cliffs at Conway Flat (Figure 1; Massey et al., 2018, 2020a; Bloom et al., 2021). Conway Flat lies c. 30 km south along the coast from the township of Kaikōura and, during the 2016 earthquake, experienced PGAs of c. 0.2 g (Figure 1; Bradley et al., 2017) with widespread cliff collapse.

## 2.2 Conway Flat Study Area

Except for a relatively small alluvial plain that surrounds the township of Kaikōura, the Northeast coast of New Zealand's South Island is generally steep and rocky. Hillslopes are primarily composed of heavily jointed Lower Cretaceous basement rocks of the Pahau Terrane and younger Upper Cretaceous to Neogene sedimentary units that are, in places, overlain by less consolidated Pleistocene alluvial and fluvial gravels (Figure 2). At Conway Flat, situated between the Conway and Waiau river mouths (Figure 2), weak Neogene Greta Formation mudstone (Uniaxial Compressive Strength <2 MPa) with massive near horizontal bedding (Rattenbury et al., 2006; Massey et al., 2018) is overlain by Pleistocene-Holocene Gilbert-style fan delta deposits that form steep coastal cliffs c. 50 to 70 m in height (McConnico and Bassett, 2007). The coastal cliffs are regularly bisected by fluvial gullies which drain the terraces at Conway Flat and portions of the nearby Hawkswood Range (Figure 2).

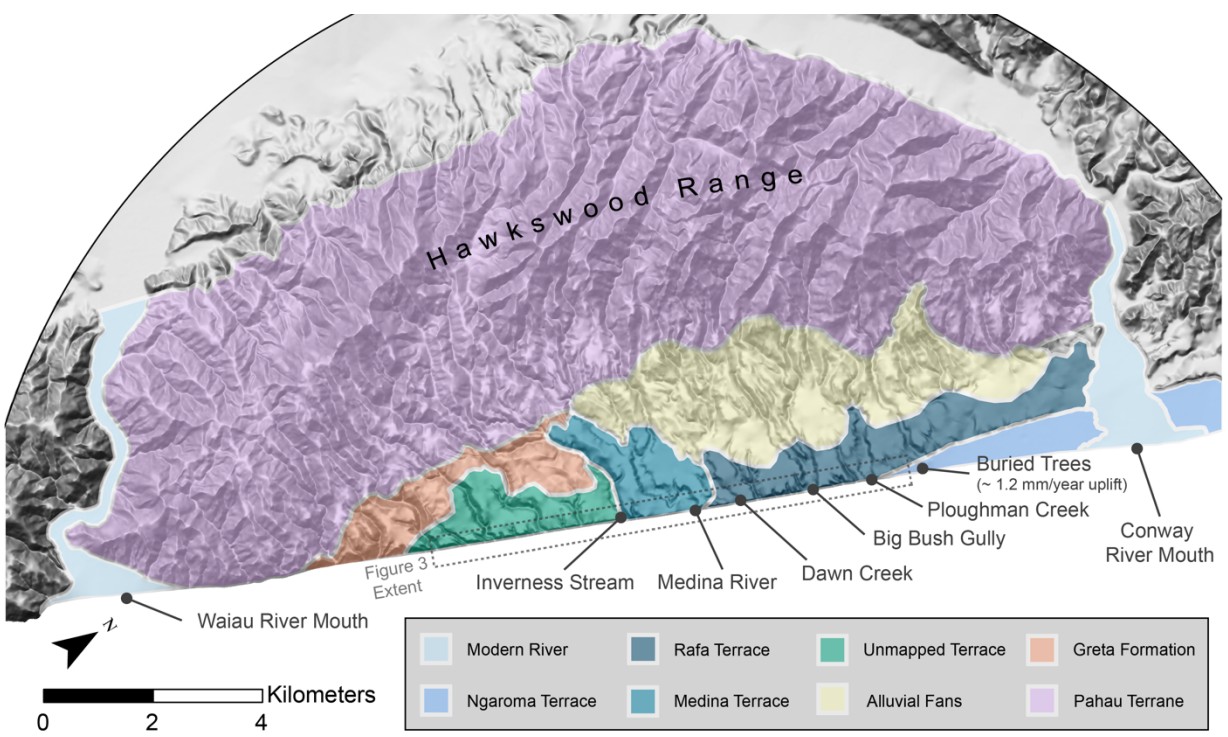

**Figure 2: Overview of the Conway Flat Coast between the Conway and Waiau river mouths. Simplified surface geology including mapped (McConnico 2012) and unmapped terraces are shown over a multidirectional hillshade derived from an 8 m DEM (LINZ, 2022). Major named streams within the study area that drain the adjacent Hawkswood Range are labelled alongside the location of buried trees used for radiocarbon dating by Ota et al. (1996) and Barrell et al. (2022). The approximate uplift rate determined by Barrell et al. (2022) from the buried trees is included in the label. The location of the study area and Figure 3 is included as a dashed grey box.**

McConnico (2012) mapped several major fan delta sequences which form terrace surfaces (Figure 2) and are present within the cliff faces at Conway Flat. The Medina fan delta forms the oldest of these mapped terraces (c. 92 to 95 ka, McConnico 2012) and makes up much of the coastal cliff face between Inverness Stream to the south and Dawn Creek to the north (Figure 2). The younger Rafa Terrace (c. 52 to 79 ka) extends north to Ploughman Creek (Figure 2, McConnico 2012) and comprises two fan delta sequences: the Dawn and Big Bush Gully fan deltas. Between Big Bush Gully and Dawn Creek, the Dawn fan delta unconformably overlies Greta Formation within the coastal cliff face; between Big Bush

Gully and Ploughman Creek the cliff face consists entirely of marine and overlying beach/fluvial facies of the Big Bush Gully fan delta. The youngest terrace surface (c. 8 ka) is formed by the Ngaroma terrace which consists of estuarine facies formed lateral to the Big Bush Gully fan delta as well as overlying fluvial and debris flow deposits (McConnico and Bassett 2007, McConnico 2012). The coastal intersection of the Ngaroma and Rafa terraces just north of Ploughman Creek forms the northern extent of our study area (Figure 2). The extent of an unmapped terrace consisting of unconsolidated sediment overlying Greta Formation to the south of Inverness Stream (Figure 2) forms the southern boundary of our study area.

While most coastal slopes in the Kaikōura region are both anthropogenically modified and buffered from direct wave action by low shore platforms and uplifted marine terraces (Mason et al., 2018; Stringer et al., 2021), the terraces and coastal cliffs at Conway Flat have had limited, to no, anthropogenic modification and are subject to direct wave action at high tide. A coarse sand and gravel beach stretches away from the cliffs at low tide. To our knowledge no previous work has been published on multidecadal coastal cliff retreat at Conway Flat.

Like much of New Zealand's tectonically active South Island, Conway Flat experiences occasional strong earthquake shaking as well as periodic heavy rain and storm surge from a combination of ex-tropical cyclones and other large storm events. Average rainfall at Conway Flat from 1949 to 2010 was 797.45 mm/year, with more rainfall occurring during the winter months from June to October (NIWA 2022). Other than during the 2016 Kaikōura earthquake, the strongest historical shaking at Conway Flat likely occurred during the 1901 Mw 6.9 Cheviot earthquake (epicenter c. 5 km from Conway Flat), the 1951 Mw 5.9 Cheviot earthquake (epicenter c. 20 km from Conway Flat), and associated aftershocks from these two events (Figure 1; GeoNet 2022, Downes and Dowrick 2014, Eiby 1968). Other strong earthquakes have occurred locally in the historic record, for example the 1965 Mw 6.1 Chatham Rise earthquake (epicenter c. 60 km from Conway Flat) or the 1987 Mw 5.2 Pegasus Bay earthquake (epicenter c. 50 km from Conway Flat), however, it does not appear that these events resulted in significant shaking intensity at Conway Flat (GeoNet 2022, Downes and Dowrick 2014, Eiby 1968). Probabilistic seismic hazard modeling (Stirling et al., 2012) suggests an c. 50-year return period for 0.2 g PGA shaking at Kaikōura (c. 35 km to the NE). Given the proximity of Kaikōura to large seismic sources from the Hope and Kekerengu faults (Langridge et al., 2016), we would expect a slightly longer 0.2 g PGA return period for Conway Flat. Ota et al. (1996) suggested c. 2 to 3 mm/year of Holocene uplift along the Conway coast based on marine terrace heights and radiocarbon ages collected from buried trees within the Ngaroma Terrace north of Ploughman Creek (Figure 2). Recent recalibration of these radiocarbon dates by Barrell et al. (2022) suggest that regional tectonic uplift is closer to 1.2 mm/year. While these rates of tectonic uplift are loosely constrained, they generally agree with

estimates of tectonic uplift (c. 0.9 to 1.2 mm/year) in marine terraces further north on the Kaikōura

peninsula (Nicol et al., 2022). Site-specific uplift rate estimates are currently poorly constrained south

of the Ngaroma terrace and Ploughman Creek.

## 3 Methods

### 3.1 2016 Kaikōura Earthquake Retreat and Historic Cliff Retreat at Conway Flat

To evaluate historic coastal cliff retreat within the study area, we produced or acquired orthoimagery

from 8 epochs of variable resolution aerial imagery (Figure 3). Images from 1950 to 1985 were

retrieved from the Land Information New Zealand (LINZ) Crown Aerial Film Archive (LINZ, 2021)

and were processed using Agisoft Metashape (additional information in Appendix A). Two additional

orthorectified images from 2004 and 2015 were retrieved from the LINZ data service (LINZ, 2022). An

orthorectified image from 2017 was sourced from Massey et al. (2020b) and, finally, in January 2022

high-resolution helicopter based aerial imagery and lidar data were collected. The available data

consists of both full and partial coverage imagery of the study area (Figure 3).

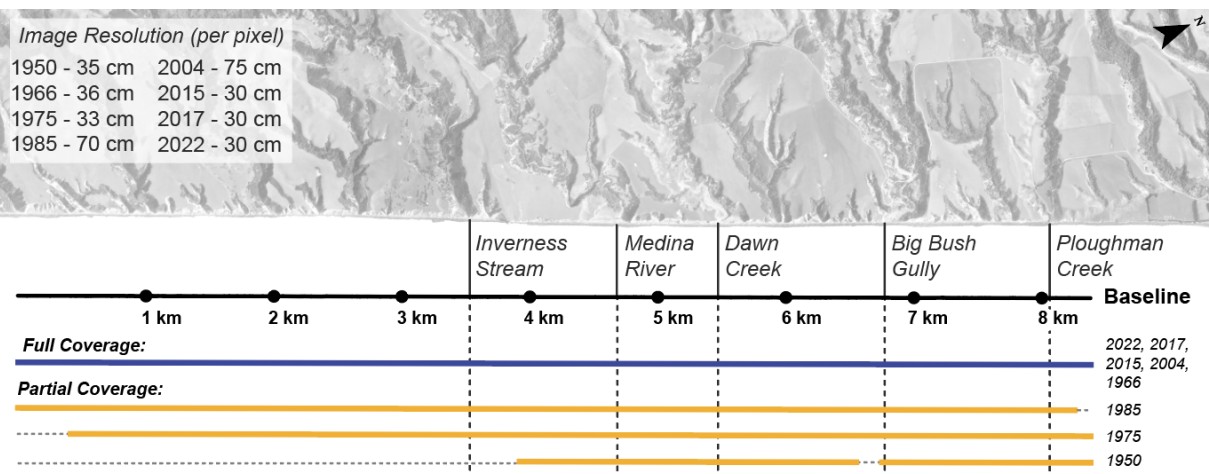

**Figure 3: Extent of Historic Aerial Imagery at Conway Flat and image resolution. Imagery with full and partial coverage of the study area at Conway Flat is shown by solid lines beneath an example of an orthorectified aerial image from 1966 (LINZ, 2021).**

**Horizontal dashed lines correspond to gaps in the aerial imagery. Vertical dashed and solid lines indicate the approximate location of major named streams within the study area.**

Displacement modeling of the 2016 Kaikōura earthquake (Hamling et al., 2017; Zinke et al., 2019),

suggests minimal coseismic and post seismic strain at Conway Flat. As such, we use well distributed

ground control points, primarily based on the corners of farm structures, stock ponds, and roads, to

horizontally register all images to a 2017 orthorectified base image. Vertical registration was relative to

a digital surface model generated from the same 2017 imagery by Massey et al. (2020b). Additional

well-spaced control points were excluded from the production of the orthoimages and were used to

evaluate georeferencing uncertainty and image distortion in each epoch of imagery. The uncertainty

between these control points was interpolated to produce an estimate of 1σ image georeferencing
uncertainty for each image set (Figure 4, additional information in Appendix A).

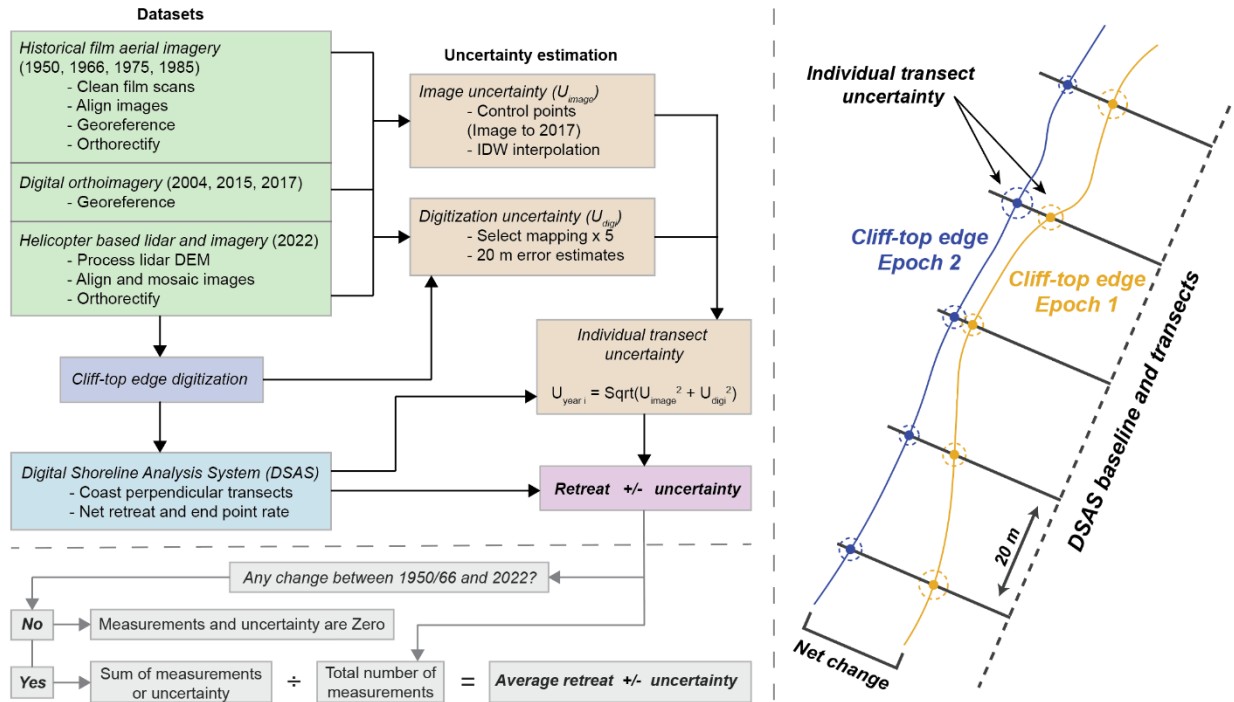

**Figure 4: Cliff-top measurement workflow and schematic.** Uncertainty was estimated and cliff-top edges were manually digitized using processed aerial image orthomosaics. Retreat and retreat rates were estimated over image time windows using the USGS
Digital Shoreline Assessment System (DSAS; Himmelstoss et al., 2021) and an estimated uncertainty was assigned to each transect. Estimates of average retreat and retreat rate include all measured transects and all unmeasured transects where no change was observed over the study time window from 1950/66 to 2022.

The upper edge of the coastal cliffs was manually mapped in each epoch of imagery and the USGS
Digital Shoreline Analysis System (DSAS; Himmelstoss et al., 2021) was used to produce
approximately perpendicular transects at 20 m intervals along the coastline (Figure 4). Using these
transects as sampling locations, we estimate both image-to-image and overall cliff retreat rates between
1950 and 2022 (Figure 4). Values of net retreat and retreat rate are reported alongside an uncertainty
which compounds image georeferencing uncertainty of the two datasets and the estimated uncertainty in
our digitization of the cliff edge. We treat this combined value as a conservative estimate of 1 σ
uncertainty (Figure 4, additional information in Appendix A). In some cases, poor image quality or gaps
in the aerial image collection made it difficult or impossible to identify a cliff edge and, in these cases,
measurements were excluded. Furthermore, in transects where dense vegetation was present across all
epochs of imagery and we were confident that no significant cliff retreat had occurred, we manually
assigned the transect a retreat rate of 0 m/year with an uncertainty of 0 m (Figure 4). Transects within
erosional gullies were excluded from our analysis as they likely represent a different erosional regime
from the majority of coastal cliff retreat at Conway Flat (Table B1).

### 3.2 2016 Earthquake Debris Volume and Post-Earthquake Debris Removal

Digital surface models (DSMs) were differenced (Figure 5) to estimate the volume of failed and
evacuated material between 2015 and 2022. We co-registered and differenced DSMs developed by
Massey et al. (2020b) using 2015 and 2017 aerial imagery to estimate the volume of material that failed
during the 2016 Kaikōura earthquake. During the 2016 Kaikōura earthquake, most cliff failures at
Conway Flat occurred as toppling or translational blockslides that transitioned into debris avalanches at
the base of the relatively geometrically-simple cliff face. As such, we assume that increases in elevation
between 2015 and 2017 that fall within the mapped extent of cliff failures from the 2016 Kaikōura
earthquake (Massey et al., 2020a) represent an accumulation of landslide debris. For each mapped
failure we multiply the sum of the gained elevation values by the area of each pixel (4 m$^2$) to estimate
an overall volume. Further, to estimate the volume loss of failed cliff material due to coastal erosion
following the 2016 Kaikōura earthquake, we co-register the 2015 DSM with a DSM developed from
high-resolution aerial lidar data collected in January 2022. Following the same method for volume
calculation as the 2015 to 2017 DSMs we estimate a remaining volume of failed material in 2022.
While we do observe some minor secondary cliff failure in the 2022 imagery, we assume that any
negative difference in the volume of debris between the 2015/2017 and 2015/2022 datasets represents
erosion of failed landslide debris following the 2016 earthquake. Our estimate therefore represents a
minimum rate of debris removal. To make a conservative estimate of 1σ uncertainty for our volume
measurements, we assume a systematic vertical offset in our DSMs based on DSM differencing outside
of the mapped landslide extents (additional information in Appendix A).

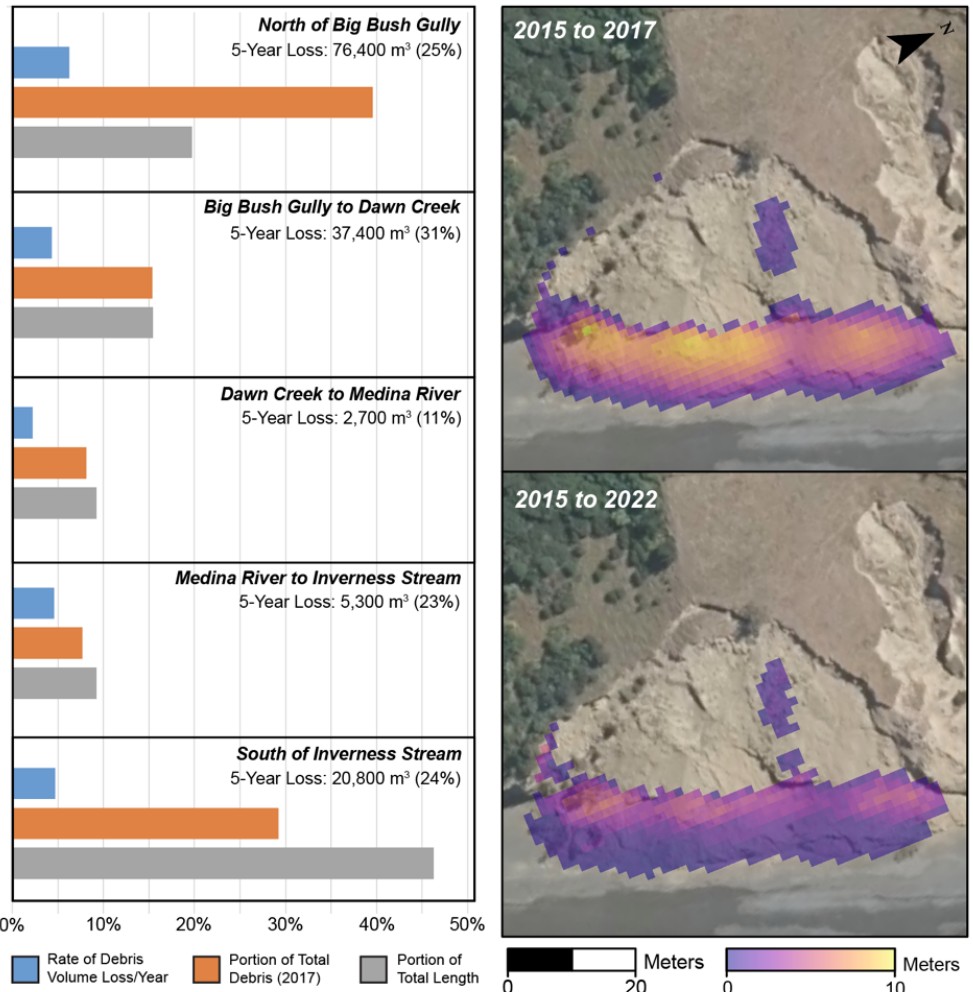

**Figure 5: Statistics on earthquake related debris volume in 2017 and 2022 and an example of measured debris volume change. For each section of the coastline, the rate of debris removal per year is plotted in blue (average: 5% per year), the portion of total earthquake related debris as seen in 2017 is plotted in orange, and the percent of the study area coastline length is plotted in grey. Assuming an even distribution of debris within the study area, the portion of total debris and the portion of study area length should be roughly equivalent within each section, however there is proportionally more debris north of Big Bush Gully and less debris south of Inverness Stream. The total amount and percentage of debris removed between 2017 and 2022 is reported for each section of the coastline. In the panels on the right, a 2017 orthomosaic (Massey et al., 2020) is overlain by an example of digital surface model differencing for 2015 and 2017 as well as 2015 and 2022. The difference in height of debris between the two time windows suggests post-earthquake debris removal from the beach.**

## 4 Results

### 4.1 Cliff Retreat from the 2016 Kaikōura Earthquake

The influence of the 2016 Kaikōura earthquake at Conway Flat is constrained by aerial imagery collected in January 2017 and January 2015 (Figure 6). Between these two image sets, we observed a maximum of c. $19.1 \pm 1.3$ m ($\pm$ combined uncertainty, $1\sigma$, Figure 4) of retreat with an average retreat of c. $3.4 \pm 1.0$ m across the study area. Of the 20 m transects that were measured between 2015 and 2017, c. 61% exhibited retreat greater than 1 m and c. 42% retreat greater than 3 m.

Retreat between 2015 and 2017 was spatially variable across the study area (Figure 6). North of Big Bush Gully, we observed, on average, c. 4.9 ± 1.3 m of cliff retreat. The coastal cliffs in this section of the study area consist almost entirely of stratified, unconsolidated to weakly consolidated, gravelly

Gilbert-style fan delta deposits of the Big Bush Gully fan delta (McConnico and Bassett 2007, McConnico 2012). Following the 2016 Kaikōura earthquake, we primarily observed large debris avalanche deposits at the base of the cliffs which appear to originate from the upper cliff edge. In several cases we also observed evidence of larger translational blockslides within the debris avalanche deposits. At the southern end of this section, just north of Big Bush Gully, there is an angular

unconformity visible within the cliff face where fan delta deposits overly Neogene age mudstone of the Greta Formation (Figure 6). During the 2016 earthquake, the Greta Formation in this lower portion of the cliff remained largely intact while the overlying unconsolidated sediment of the Big Bush Gully fan delta appears to have failed as a debris avalanche (Figure 6).

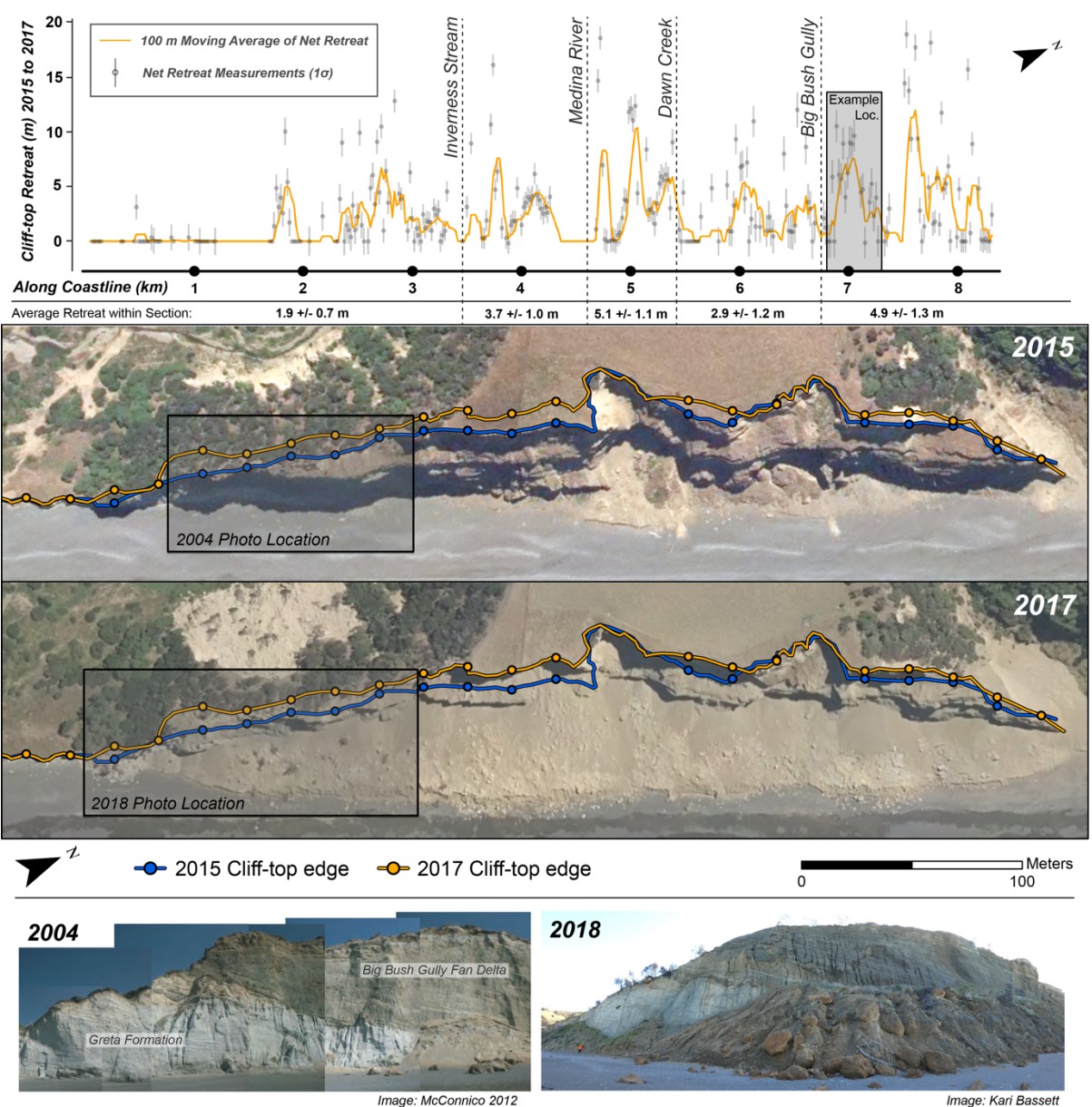

**Figure 6: Measured cliff retreat between 2015 and 2017 and examples of coastal cliff failure. In the first panel, cliff-top retreat is plotted against distance along the study area baseline (Figure 3). Individual measurements are shown as grey points with error bars representing estimated uncertainty for the given measurement. A 100 m moving average of the data, which assumes that**

measurement gaps are zero, is plotted as an orange line. The grey inset north of Big Bush Gully identifies the location of the aerial images from 2015 (LINZ, 2022) and 2017 (Massey et al., 2020) in the next panel. In the aerial image panel, blue lines and associated measurement points represent the 2015 cliff edge while orange lines and points represent the 2017 cliff edge. The location of representative photos in the next panel are identified by black boxes in the 2015 and 2017 aerial images. Photos in the final panel show an example of the coastal cliffs at Conway Flat before and after the 2016 Kaikōura earthquake.

To the south of Big Bush Gully, between Big Bush Gully and Dawn Creek, we observed on average c. 2.9 ± 1.2 m of cliff retreat (Figure 6). In most of this section of coastline, Dawn fan delta deposits unconformably overlie weakly lithified mudstone of the Greta Formation. During the 2016 earthquake, many failures occurred as debris avalanches sourced from the overlying fan delta material. In most cases, it appears that the Greta Formation did not fail beneath the terrace material. We do observe

several isolated instances where potentially pre-existing rotational and translational failures occur within the underlying Greta Formation and these may have facilitated additional back wasting of the upper cliff face and cliff-top edge. Further site-site specific investigation beyond the scope of this work would be required to further elucidate the relative contribution of the Greta Formation to historical failures at Conway Flat.


Moving south from Dawn Creek to the Medina River, we observed an average retreat of c. 5.1 ± 1.1 m, the highest average cliff retreat within the 2015 to 2017 time window (Figure 6). The cliff face in this section of coastline is entirely made up of Dawn fan delta deposits. We observed a small terrace within the upper third of the slope where slightly less indurated sediment, similar to the material observed

further north, overlies more indurated fan delta deposits. During the 2016 Kaikōura earthquake most failures in this section of the study area occurred as debris avalanches from the upper cliff face above this terrace.

From the Medina River to Inverness Stream, we observed, on average, c. 3.7 ± 1.0 m of cliff retreat.

The cliff in this section of coastline is primarily composed of older and more consolidated Medina fan delta deposit which has experienced significantly less retreat over the past 72 years. We observed several large rock falls and some smaller debris avalanches from the upper cliff resulting from the 2016 earthquake however these failures were more isolated than the widespread failures to the north.

Finally, south of Inverness Stream and the mapped extent of the Medina Terrace (Figure 2), we observed, on average, c. 1.9 ± 0.7 m of retreat (Figure 6). Here, Greta formation mudstone forms a terraced cliff face that likely buffers the overlying variably thick package of unmapped unconsolidated sediment (and the upper cliff edge) from wave-driven erosion. During the 2016 earthquake, we observed significant debris avalanching from the overlying unconsolidated sediment in this section of

coastline but little change in the position of the lower cliff face.

Following the 2016 earthquake, between 2017 and 2022, we did observe some local cliff retreat and additional rockfall that may be related to earthquake aftershocks, however, on average, retreat was relatively low (c. 0.4 m) and fell within the uncertainty of our measurements (c. ± 1.5 m).

**4.2 Debris Volume and Post-Earthquake Debris Removal**

Between 2015 and 2017, we estimate that c. 302,100 ± 86,600 m³ (± 1σ) of material failed along the 8 km of the Conway Flat coastal cliffs (Figure 5). As of January 2022, the total volume of failed material remaining on the beach from these same failures was c. 225,700 ± 93,300 m³, a net loss of c. 25% of earthquake-related failed material within 5 years. This estimate includes c. 22,700 ± 6,200 m³ of debris that was added between 2017 and 2022. North of Big Bush Gully, we observe a higher rate of debris removal where c. 31% of earthquake related debris was evacuated between 2017 and 2022 (Figure 5).

**4.3 Historic Cliff Retreat at Conway Flat**

The study area was first captured by full aerial imagery in 1966 (Figure 3) and the average retreat rate over the entire area was c. 0.16 ± 0.04 m/year from 1966 to 2022 (Figure 7). The study area north of Inverness Stream, captured in earlier aerial imagery from 1950 (Figure 3 and 7), had an average retreat rate of c. 0.25 ± 0.03 m/year (1950 and 2022) with a maximum retreat of c. 61.5 ± 2.2 m. Prior to the 2016 Kaikōura earthquake, the average overall retreat rate for the entire study area was c. 0.11 ± 0.04 m/year (1966 to 2015) and the retreat rate north of Inverness Stream was c. 0.14 ± 0.04 m/year (1966 to 2015) or c. 0.2 ± 0.03 m/year (1950 to 2015).

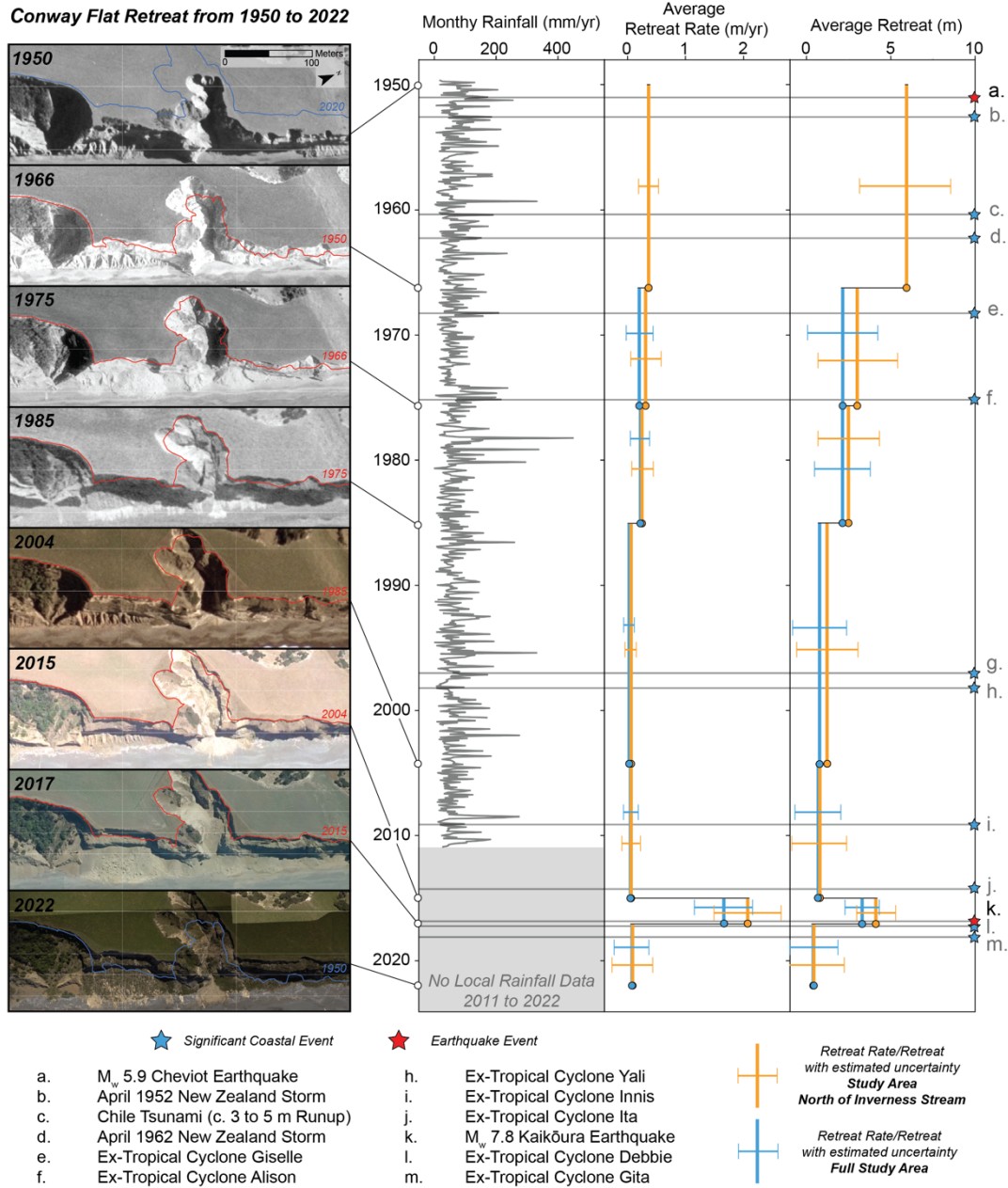

**Figure 7: Examples of cliff retreat at Conway Flat, monthly rainfall in study area, retreat rate, retreat, and significant events from 1950 to 2022. Within each example image (LINZ, 2021; 2022), the cliff edge from the previous image is shown by a red line. Blue lines in the first and last example image represent the total retreat between 1950 and 2022. White points and black lines connect the respective images to the timeline on the right. Monthly rainfall totals from a rain gauge measured daily at Conway Flat between 1949 and 2010 (NIWA, 2022), are plotted as a grey line in the first plot on the left. The average retreat rate (middle plot) and average retreat between time windows (right plot) are presented as vertical lines alongside estimated uncertainty indicated by horizontal error bars. Orange lines represent the study area north of Inverness Stream (1950 to 2022) and blue lines represent the entire study area (1966 to 2022). Significant coastal events (blue stars) and earthquakes (red stars) at Conway Flat between 1950 and 2022 are plotted along the far-right edge and are connected to the timeline by grey lines.**

### 4.3.1 Temporal Variability of Historical Cliff Retreat

The historical cliff retreat rate at Conway Flat was variable between time windows (Figure 7). On average, we observed widespread cliff retreat between 1950 and 1966 (c. 0.38 ± 0.17 m/year) and 2015 and 2017. We observed more localized cliff retreat between 1966 and 1975 (c. 0.23 ± 0.23 m/year) and 1975 to 1985 (c. 0.24 ± 0.16 m/year). Average retreat between 1985 and 2004 (c. 0.04 ± 0.08 m/year),

2004 and 2015 (0.07 ± 0.12 m/year), and 2017 and 2022 (0.09 ± 0.3 m/year) fell within the estimated uncertainty of their respective image sets. Some changes in local cliff position were evidenced by failure scars and debris piles within each of these time windows but we do not observe widespread change in cliff-top position.

### 4.3.2 Spatial Variability of Historical Cliff Retreat

Across individual coastline transects, retreat rates at Conway Flat ranged from 0 to 0.86 ± 0.03 m/year over the full-time window (Figure B1). We observed the highest overall retreat rates (on average c. 0.29 ± 0.05 m/year from 1966 to 2022) in the northernmost portion of the study area north of Big Bush Creek, Figure B1) with retreat rates decreasing toward the south. The lowest average retreat was observed south of Inverness Stream (c. 0.09 m ± 0.03 m/year from 1966 to 2022). These observations south of Inverness Stream correlated well with an increasing density of vegetation on the cliff face that may be indicative of longer-term coastline stability.

## 5 Discussion

### 5.1 Cliff Retreat and Lithology

Underlying geology appears to largely govern the spatial variability of coastal cliff retreat at Conway Flat over the historical record. Where the cliff face consisted entirely of unconsolidated fan delta deposits, for example in the Big Bush Gully fan delta north of Big Bush Gully, we observed more substantial historical retreat. Where Greta Formation mudstone or more indurated fan delta deposits like those of the Medina fan delta were present in the lower cliff face, in general, we observed lower retreat rates. In the Kaikōura region and across New Zealand, failures in tertiary sediment including the Greta Formation mudstone tend to occur as large planar slides often failing along preferentially oriented bedding planes (Pettinga, 1987; Mountjoy and Pettinga, 2006; Singeisen et al., 2022) or as shallow debris avalanches in more weathered sections of the rock mass (Massey et al., 2018). We do not observe evidence of planar sliding at Conway Flat over the historical record and most retreat of the underlying Greta formation appears to result from a combination of shallow debris avalanching, observed in some aerial imagery, and more gradual erosion due to wave action. Determining the extent to which failure mechanisms within different facies of the fan deltas and Greta Formation govern historical cliff-top retreat at Conway Flat is largely beyond the scope of this study; however it does appear that more indurated material (with assumed higher shear strength) in the lower cliff face may buffer the upper cliff face from wave action effectively reducing the non-seismic rate of cliff-top retreat (Emery and Kuhn 1982). For our purposes here, we define the non-seismic rate of cliff-top retreat as retreat from any non-seismic source. This may include, but is not

limited to, failure of the cliff-top during rainfall events, failure from undercutting of the cliff face, and/or weathering and background gravitational failure.

Most failures from the 2016 Kaikōura earthquake occurred as debris avalanches from the upper cliff face with very little retreat of the lower cliff face. While the long-term position of the coastal cliffs at Conway Flat may be governed by wave processes undercutting the lower cliff face, earthquakes may disproportionately influence cliff-top retreat over multiple decades through topographic amplification of strong ground motion in the upper cliff face (e.g. Ashford et al., 1997; Massey et al., 2022).

In addition to lithology, several other factors may further influence the rate of local cliff retreat at Conway Flat over the historical record but these are more challenging to quantify. For example, the rate of tectonic uplift and sediment transport may influence beach height in relation to the base of the cliffs at Conway Flat (e.g., Horton et al., 2022) but it is not possible to quantify these changes across our historical image

datasets. Likewise, local aspect of the cliff face in relation to variable incoming wave direction may influence the rate of cliff retreat but information on changes in wave direction through time are unavailable.

## 5.2 Post-Earthquake Sediment Loss

The efficient evacuation of failed material at Conway Flat makes it difficult to identify the historical source of failures. Assuming a steady c. 15,300 $m^3$/year (c. 5%) annual rate of debris removal from the base of the cliffs at Conway Flat, as we observed in the 5 years following the 2016 Kaikōura earthquake, we expect that nearly all earthquake related debris will be removed within c. 20 years of the earthquake. Rates of volume loss appear to vary slightly based on the composition of debris with higher-than-average

rates of debris removal north of Big Bush Gully where debris consists of largely unconsolidated fan delta deposits and much lower than average rates of debris removal between Dawn Creek and the Medina River where deposits consist of more intact blocks which we infer are from higher shear strength facies of the fan delta deposit (Figure 5). The extent to which storm surge from events like Ex-tropical Cyclone Gita (Figure 7) and variability in longshore sediment transport (Larson and Kraus, 1993; Dickson et al., 2007;

Karunarathna et al., 2014) influence the removal of failed debris at Conway Flat remains largely unclear due to our limited number of image epochs but it is possible that such events modulate the rate of debris removal over time.

Interestingly, in imagery from 1966, we observed almost no material at the base of the coastal cliffs

despite an average cliff-top retreat of c. 5.9 m between 1950 and 1966 (Figure 7). Applying the same rate of debris removal in the 5 years following the 2016 Kaikōura earthquake to the 1950 to 1966 time window

does not fully explain the lack of debris in 1966. Assuming failures occurred early in the time window, a number of large storm events alongside a c. 3 m run-up tsunami associated with the 1960 Chile Earthquake (Figure 7) may have increased the rate of debris removal.

## 5.3 Cliff Retreat and Earthquake Shaking

Over long timescales (i.e. longer than the historical record), the rate of coastal erosion at the base of the Conway Flat cliffs may limit the extent of cliff-top retreat and, in turn, the long-term influence of subaerial triggers like earthquakes or rainfall. This is because some oversteepening of the cliff face is likely a prerequisite for cliff-top failure (Wolters and Müller 2008). Prior to 2016, most of the cliff face at Conway Flat was near-vertical in many places (Figure 6), an indication of dominant marine erosion (Emery and Kuhn 1982). However, over multidecadal timescales, the rate of cliff-top and base retreat may vary substantially due to the greater temporal variability of subaerial triggers.

We observe this variability in the historic record of cliff retreat at Conway over the past 72 years. Direct observational evidence suggests that c. 24% of 72-year retreat (in the 2015 to 2017 time window) resulted from the 2016 Kaikōura earthquake. We hypothesize that over multiple decades, large subaerial landslide triggering events, for example earthquakes or storms, contribute disproportionately to cliff-top retreat at Conway Flat while coastal erosion dominates retreat at the base of the cliffs, in turn creating a steeper cliff face more susceptible to subaerial triggers. The historical analysis here excluded areas with clear evidence of fluvial incision, for example rills and gullies, and, as a result, there is likely a limited influence of surface run-off. Furthermore, Conway Flat has seen little anthropogenic or other biologic change over the study period that could significantly influence the rate of cliff retreat. Observations of cliff face hydrology are limited with the exception of vegetation and some minor seeps.

Large rainfall events and storms could explain the temporal variability in cliff-top retreat at Conway Flat but the historic record of these events has little correlation with the observed retreat rate (Figure 7). On the other hand, the nearby 1951 $M_w$ 5.9 Cheviot Earthquake with its six $M_w$ 5.0+ aftershocks provides a plausible explanation for significant retreat observed between 1950 and 1966. Although there is no direct evidence of coastal cliff failures at Conway Flat from the 1951 Cheviot earthquakes, two lines of evidence lend credibility to their contribution. First, regional documentation of shaking and damage from the 1951 Cheviot earthquake main shock suggests a Modified Mercalli (MM) Intensity of VI to VII at Conway Flat (Downes and Dowrick, 2014), similar to shaking intensity from the 2016 Kaikōura earthquake. An implementation of the ShakeMapNZ model (Horspool et al., 2015) using historic observed ground motion data as well as the damage and felt-reports from Downes and Dowrick (2014) suggests that the 1951 Cheviot earthquake produced ground motion with a PGA between c. 0.1 and 0.2 g at Conway Flat (Figure 1), very similar to the modeled PGA from the 2016 Kaikōura earthquake (Bradley et al., 2017; Figure 1).

Second, as discussed previously, applying the rate of debris removal following the Kaikōura earthquake to the 1950 to 1966 time window does not fully explain a lack of debris in 1966. Assuming little change in the rate of beach erosion between time windows, this suggests that failures likely occurred early in the 1950 to 1966 time window exposing debris to a number of intense storm surges and tsunami inundation in 1960 that may have enhanced debris removal (Figure 7).

Together, the 1951 and 2016 earthquakes account for a significant portion of the overall retreat at Conway Flat in the past 72 years. Excluding both earthquakes from the historical estimate of cliff retreat at Conway Flat north of Inverness Stream, reduces the retreat rate to c. 0.14 m/year or c. 56% of the total (0.25 m/year) retreat over the past 72 years. This being said, given the relatively short return interval of sufficient shaking to induce cliff retreat at Conway Flat over the historical record (c. 58 years considering the last three earthquakes from 1901, 1951, and 2016), including both the Kaikōura and Cheviot earthquakes could overestimate the multidecadal cliff-top retreat rate.

Excluding either the 2015 to 2017 time window including the 2016 Kaikōura earthquake or the 1950 to 1966 time window including the 1951 Cheviot earthquakes from our estimates results in c. 0.16 to 0.20 m/year of cliff-top retreat at Conway Flat. These values represent a best estimate of the multidecadal cliff-top retreat rate at Conway Flat over the historical record and are, on average, c. 45% greater than the estimated non-seismic retreat rate. Following the 2016 earthquake we observed a steep upper cliff face, likely still susceptible to failure, across much of Conway Flat (Figure 6). Given relatively low ground motion at Conway Flat during both the 2016 Kaikōura earthquake and the 1951 Cheviot earthquake, it remains possible that stronger ground motion could result in greater single event retreat. In this case, strong ground motion could have an even larger influence on multidecadal cliff-top retreat.

## 5.4 Implications for Multidecadal Cliff Retreat Estimates in Tectonically Active Regions

In tectonically active regions that have not experienced a sufficiently large earthquake in the historic record, excluding earthquake contributions will result in underestimates of multidecadal coastal cliff retreat over multiple earthquakes. While in the case of Conway Flat this underestimation is likely around 30%, the degree to which the historical record underestimates multidecadal cliff-top retreat at any given site will be heavily modulated by several factors. These factors include site characteristics like cliff height, lithology and slope that influence cliff susceptibility to earthquake induced failure (Massey et al., 2022), the overall rate of cliff-top retreat as compared with retreat during a single earthquake event, and the expected return intervals, magnitudes, durations and frequency content of earthquake shaking. Simplified, these factors fall into three primary categories: (*a*) magnitude of single-event cliff-top retreat related to ground motion, (*b*) non-seismic cliff-top retreat rate, and (*c*) return interval of sufficient ground motion to result in single-event retreat.

Using these three inputs, and assuming a cliff-top susceptible to failure, a simple equation can be defined to determine the influence of earthquakes on the multidecadal rate of coastal cliff-top retreat ($Y$) as follows Eq. (1):

$$Y = \frac{a}{c} + b, \qquad (1)$$

Effectively, the multidecadal coastal cliff-top retreat rate equals the sum of earthquake related cliff-top retreat and the total retreat from other non-seismic processes over the return period of sufficient shaking.

Based on our observations of the 2016 Kaikōura earthquake, we estimate that strong ground motion around PGA > 0.1 g is sufficient to produce cliff-top retreat at Conway Flat though it is possible that even stronger ground motion could result in greater retreat. The historic record of earthquakes at Conway Flat suggests that sufficient ground motion to induce cliff-top retreat occurs approximately every 50 years. While seismic hazard is lower (and thus return period is longer) at Conway Flat, the historical record is consistent with seismic hazard curves for Kaikōura which suggest a 50-year return period for PGAs of 0.2 g (Stirling et al., 2012). If we assume that the c. 0.11 m/year retreat rate we measured for the entire study area at Conway Flat from 1966 to 2015 represents the non-seismic coastal cliff-top retreat rate and that the coastal cliff-top retreats an average of 4 m each time there is sufficiently strong ground motion (c. every 50 years), we can apply the equation above to estimate a multidecadal coastal cliff retreat rate of c. 0.19 m/year (Figure 8). This value is similar to our historical estimate of multidecadal cliff retreat at Conway Flat (see Section 5.3).

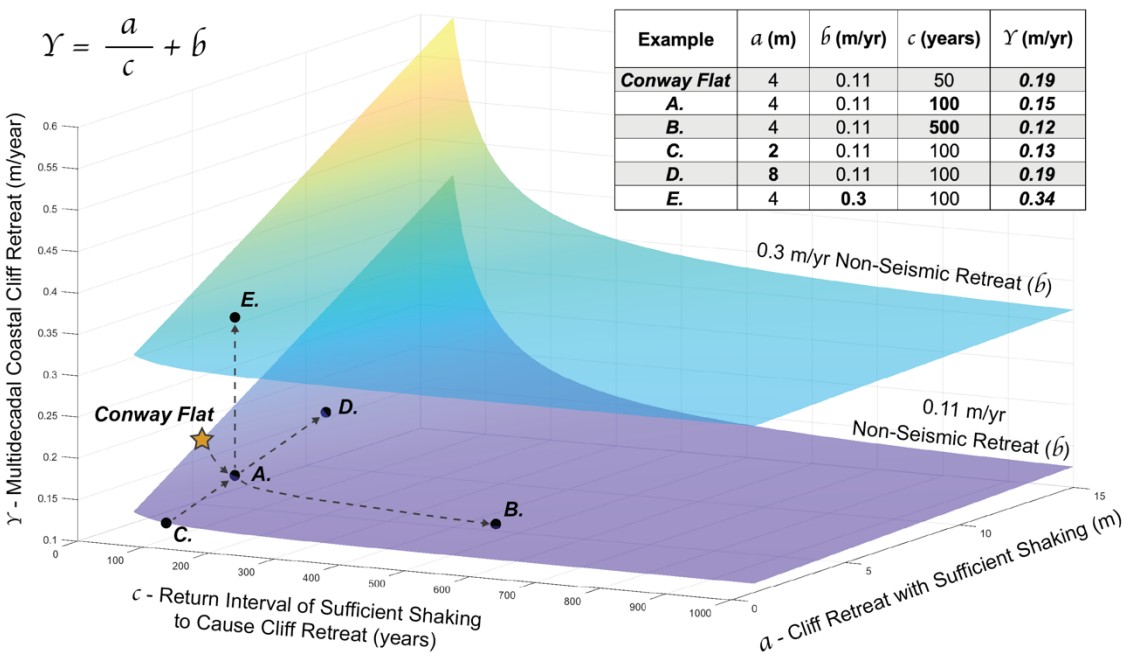

**Figure 8: Plot showing the three primary factors controlling earthquake influence on multidecadal coastal cliff retreat: amount of cliff retreat from sufficient earthquake shaking (a), the rate of non-seismic cliff retreat (b) and return interval of sufficient earthquake shaking to cause cliff retreat (c). Increasing return interval results in a hyperbolic decay of the multidecadal coastal cliff retreat rate (A. to B.) while multidecadal coastal cliff retreat varies linearly with shaking related cliff retreat (C. to D.) and non-seismic retreat (A. to E.).**

By varying the inputs involved in Equation 1, we can further explore how sufficiently strong ground motion may influence the multidecadal rate of cliff retreat at other sites (Figure 8). The recurrence of sufficient ground motion for cliff retreat ($c$) forms a hyperbola where longer return intervals result in significantly lower single-event influence on multidecadal retreat (Figure 8). Using the same example from Conway Flat above, increasing the return interval to 100 years results in a multidecadal cliff retreat rate of 0.15 m/year (0.04 m/year higher than non-seismic retreat alone, Figure 8, A.) while increasing the return interval to 500 years results in a multidecadal cliff retreat rate of c. 0.12 m/year (0.01 m/year higher than non-seismic retreat, Figure 8, B.). Varying the amount of cliff retreat from earthquake shaking ($c$) or the non-seismic retreat rate ($b$) linearly scales the influence of ground motion (Figure 8, C.-E.). The relative influence of earthquake related retreat at any given site is directly proportional to the non-seismic retreat rate. For example, at a site with a high non-seismic retreat rate but a relatively low single-event earthquake retreat (Figure 8, E.), shaking is unlikely to have a strong influence on the multidecadal retreat rate over multiple sufficiently large earthquakes. Alternatively, at a site with a relatively low non-seismic retreat rate and a relatively high single-event earthquake retreat (Figure 8, D.), earthquakes could have a substantial influence on the multidecadal retreat rate at short return intervals of sufficient shaking.

North of the study area at Conway Flat, the low-lying coastal cliffs of the Ngaroma Terrace (Figure 2) experienced very little coseismic failure in 2016 despite similar material and rate of non-seismic coastal erosion (c. 0.2 m/year between 1950 and 2017). This may be a result of varying site response to ground motion and underlines the challenge in making generalities across coastal cliffs, especially in regions with different lithologic and topographic site conditions. That being said, Conway Flat experienced widespread cliff retreat from relatively moderate ground motion and should serve as an important demonstration of the potential for historical rates of coastal cliff retreat to significantly underestimate multidecadal retreat over multiple earthquakes. In regions like coastal California where high population exposure to steep coastal cliffs coincides with frequent earthquake shaking (e.g. Griggs and Plant 1997), understanding how earthquakes influence the multidecadal retreat of coastal cliffs could be important for calibrating effective forecast models. Geomorphic evidence of past earthquake events may not be preserved, even over historical timescales, so investigations may need to integrate seismic hazard analysis, geotechnical site characterization, physics-based modeling of coastal cliff response to earthquake shaking, and regional earthquake-induced landslide susceptibility analysis.

## 6 Conclusions

With the rate of coastal cliff retreat set to increase due to climate change induced sea level rise, accurately modeling and forecasting future cliff retreat is extremely important, particularly in areas with

high population exposure to coastal hazards. The 2016 $M_w$ 7.8 Kaikōura earthquake on the South Island of New Zealand, resulted in significant coastal cliff retreat in the area of Conway Flat where modelled ground motion was around 0.2 g PGA. We used Conway Flat as a natural laboratory to examine how earthquake shaking influences the historical record of cliff retreat. Retreat was spatially and temporally variable over the past 72 years and large earthquake induced landslide triggering events appear to disproportionately contribute to an average 0.25 m/year retreat at Conway Flat. The 2016 Kaikōura earthquake alone represents c. 24% of the total retreat over the past 72 years. Together with observations of significant retreat between 1950 and 1966, which likely resulted from the 1951 $M_w$ 5.9 Cheviot Earthquake, we estimate that earthquakes increase the multidecadal cliff retreat rate at Conway Flat by c. 45% over estimates that exclude earthquakes. Evidence of widespread failure, including failed debris, has been quickly removed by coastal erosion following the 2016 Kaikōura earthquake with an estimated c. 15,300 $m^3$ or 5% of landslide debris removed each year in the 5 years following the earthquake. In tectonically active regions that have not experienced recent earthquake related cliff retreat, the extent to which the historical record underestimates multidecadal retreat rate is highly dependent on the magnitude of coseismic and non-seismic cliff-top retreat and the return interval of sufficient ground motion to induce failure. Seismic hazard models and dynamic physical models of coastal cliffs may thus serve as useful tools for estimating the potential multidecadal influence of earthquakes on coastal cliff retreat rates.

**Appendix A – Additional Methods and Data**

**Orthomosaic processing from scanned images**

The aerial image orthomosaics discussed in the manuscript from 1950, 1966, 1975, and 1985 were processed from original digital scans in the LINZ Crown Aerial Film archive (LINZ, 2021) using Agisoft Metashape 1.8.2. The fringe of the original image scans includes fiducial marks and information on the camera lens that were matched with camera calibration certificates provided by the Crown Aerial Film archive. In Metashape, we define the fiducials for each scan before masking the image fringe.

We followed the typical Agisoft Metashape workflow as outlined in the program documentation for producing orthomosaics. Images were aligned at low quality and well distributed ground control points, sourced from a 2017 orthomosaic and derived digital surface model (DSM; Massey et al., 2020), were assigned manually to each image. Images were then realigned and optimized. We built a high-quality dense point cloud with moderate depth filtering and a height field mesh using the dense point cloud and a high face count. A geographic orthomosaic was produced from the mesh and exported to .tif format.

**Uncertainty estimation**

**Georeferencing uncertainty**

Georeferencing error and distortion result in variable uncertainty across the orthomosaics we produced from scanned aerial imagery (LINZ, 2021), those that we obtained from the LINZ Data Service (LINZ, 2022), and those that we produced from lidar. As we are directly comparing these datasets, it is

important that we characterize this uncertainty. For each orthomosaic, we identified distributed control points along the coastline (our main area of interest) and matched these control points with our base 2017 orthomosaic. We estimate a Euclidean distance between the matched control points and assign this distance as the uncertainty at each point. Because there is not a consistent uncertainty across our images, we applied inverse distance weighted interpolation in ArcGIS to interpolate our uncertainty as a

25 m/pixel continuous grid across the study area (Figure A1).

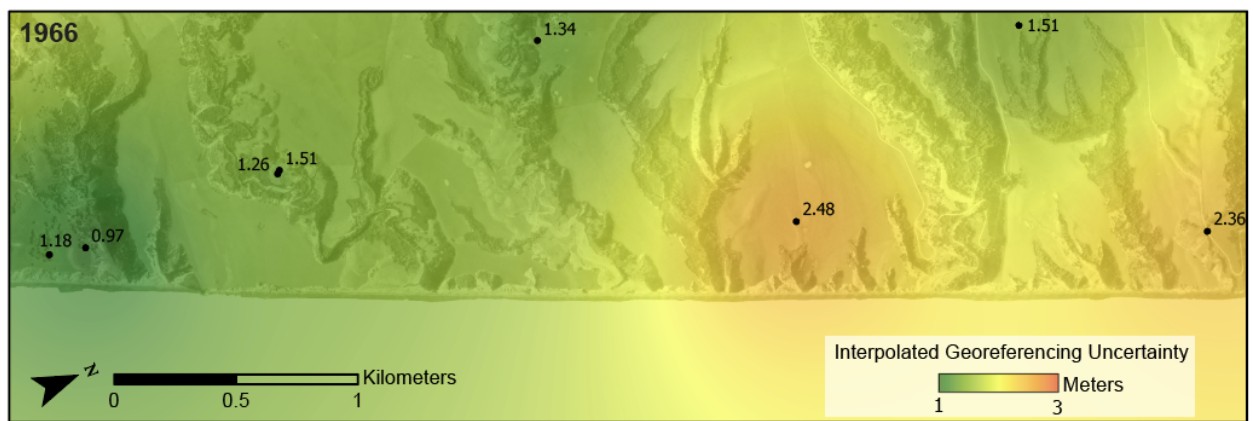

Figure A1. Example of interpolated georeferencing uncertainty in 1966 orthomosaic (LINZ, 2021). Black points with labels represent the distance between a control point and the base 2017 orthomosaic (Massey et al., 2020). IDW interpolation is used to
create a 25 m/pixel continuous grid of uncertainty across the image.

We extract the estimated georeferencing uncertainty from the interpolation at each point where a transect crosses the digitized shoreline in that image. In our case, each transect has two points where it crosses, one associated with the older image and one associated with the younger image. Georeferencing uncertainty is compounded between the two points to provide a total georeferencing

uncertainty for each transect using the following equation Eq. (A1):

$$Transect\ Georeferencing\ Uncertainty = \sqrt{U_{image\ 1}^2 + U_{image\ 2}^2} \qquad (A1)$$

where $U_{image\ 1}$ is the georeferencing uncertainty for the older cliff edge and $U_{image\ 2}$ is the georeferencing uncertainty for the younger cliff edge. As shown in Manuscript Figure 4, this georeferencing uncertainty is combined with digitization uncertainty to define an overall measurement uncertainty.

**Digitization uncertainty**

Using the same equation as we use for georeferencing uncertainty, we compound a digitization uncertainty for each transect. Unlike the georeferencing uncertainty, however, we assume that

digitization uncertainty is consistent across the image set. We conducted a blind resampling of our digitization on a representative section of coastline (the example in manuscript Figure 7) to determine

the digitization uncertainty. The same person who digitized the cliff-top across the image retraced the section of cliff edge 5 times and estimated the maximum coast perpendicular distance between all possible pairs of cliff edge traces in regular 20 m intervals along the coast. The digitization error for the image is defined as the average of these maximum distances (Table A1). This value may be greater than $1\sigma$ uncertainty, but we treat it as a conservative estimate of $1\sigma$.


**Table A1. Digitization Uncertainty for each orthomosaic (LINZ, 2021; 2022)**

| Year | Resolution | Uncertainty |
|------|-----------|-------------|
| **1950** | 0.35 | 0.69 |
| **1966** | 0.36 | 0.65 |
| **1975** | 0.33 | 0.64 |
| **1985** | 0.7 | 0.85 |
| **2004** | 0.75 | 0.88 |
| **2015** | 0.3 | 0.62 |
| **2017** | 0.3 | 0.62 |
| **2022** | 0.3 | 0.62 |

**Debris volume uncertainty**

To estimate uncertainty for our volume estimates, we assume that the mean difference between the measured DSMs, outside of the extent of mapped landslides, represents a systematic vertical offset

between the two datasets. We add this estimated vertical offset (0.72 m 2015 to 2017 and 0.92 m 2015 to 2022) to the elevation difference within the extent of mapped landslides and estimate a $+1\sigma$ debris volume. Similarly, by subtracting the estimated vertical offset, we estimate a $-1\sigma$ debris volume. Subtracting the $+1\sigma$ debris volume from the measured debris volume results in a conservative estimate of $1\sigma$ uncertainty. In reality, $1\sigma$ may be smaller as much of the area within mapped landslides was

unvegetated before and after the earthquake while much of the area outside mapped landslides was vegetated.

**Transect Locations at Conway Flat**

Transects at Conway Flat were grouped into five sections based on lithologic domains described in the manuscript. Transects are located every 20 m along a baseline which extends from south to north

(Lat/Lon -42.723591 173.408607 to -42.654044 173.446673). Transects from 0 to 3,340 m along baseline are considered to be south of Inverness Stream. Transects from 3,360 to 4,680 m along baseline fall between Inverness Stream and the Medina River. Transects between 4,700 and 5,440 m along baseline fall between the Medina River and Dawn Creek. Transects between 5,460 and 6,760 m

along baseline fall between Dawn Creek and Big Bush Gully. Finally, transects from 6,780 to 8,320 m along baseline are considered to be north of Big Bush Gully.

## Appendix B – Additional Results

### Gullies

As discussed in the manuscript, we removed gullies from our estimates of average retreat and retreat rate at Conway Flat as they likely represent different erosional mechanisms from the majority of the coastal cliff face. Here we present the average retreat rate within the gullies as they compare to the cliff retreat reported within the manuscript (Table B1).

**Table B1. Gully Statistics**

| Years | Number of Gully Transects | Average Cliff Retreat with Gullies (m) | Average Cliff Retreat without Gullies (m) | Average Gully Retreat (m) | Percent Difference between Gully and Cliff Retreat |
|---|---|---|---|---|---|
| 2017 to 2022 | 26 | 0.45 | 0.43 | 0.58 | 33.95% |
| 2015 to 2017 | 25 | 3.21 | 3.35 | 1.88 | -43.93% |
| 2004 to 2015 | 23 | 0.92 | 0.72 | 2.39 | 230.57% |
| 1985 to 2004 | 24 | 1.02 | 0.85 | 2.53 | 199.05% |
| 1975 to 1985 | 25 | 2.17 | 2.19 | 1.97 | -9.88% |
| 1966 to 1975 | 26 | 2.20 | 2.19 | 2.33 | 6.54% |
| 1950 to 1966 | 13 | 6.25 | 5.91 | 9.91 | 67.60% |
| 1966 to 2022 | 26 | 8.88 | 8.66 | 11.12 | 28.42% |
| 1966 to 2015 | 25 | 5.61 | 5.35 | 8.25 | 54.21% |
| 1950 to 2015 | 13 | 13.21 | 12.64 | 18.10 | 43.22% |
| 1950 to 2022 | 13 | 18.18 | 17.57 | 24.76 | 40.90% |

In general, gullies exhibited greater retreat than the overall cliff face. This was particularly pronounced in the 2004 to 2015 and the 1985 to 2004 time windows where gully retreat was c. 2 times as great as general cliff retreat. While it is possible that higher gully retreat, particularly in these two time windows, is related to the greater susceptibility of gullies to fluvial incision, the extremely limited number of gully transects (13 to 26) casts doubt on the reliability of a direct comparison with the general cliff transects.

### Spatial variability in historical retreat at Conway Flat

As discussed in the manuscript and demonstrated by Figure B1 there is significant spatial variability in historical retreat at Conway Flat. In general, retreat decreases north to south with the highest retreat north of Big Bush Gully and the lowest retreat south of Inverness Stream. In some sections of the Conway Flat coast, for example south of Inverness stream, lower historical cliff-top retreat could be a result of more resistant geology at the toe of the cliff that prevents undercutting of the upper cliff by wave action. Over multidecadal timescales this may result in subaerial triggers like

earthquakes playing a more important role in cliff-top retreat than wave action. When cliff-top retreat
outpaces retreat at the toe of the cliff, the steepness of the cliff-top can reduce, effectively slowing cliff-
top retreat but this is not always the case at Conway Flat. Figure B2 provides comparative profiles
demonstrating this. In profile A to A' in Figure B2 the cliff-top is influenced by both wave action at the
toe of the slope (pre-2016) and subaerial triggers at the top of the slope. Alternatively in profile B to B',
terracing occurs with Greta formation buffering cliff-top retreat from undercutting by wave action prior
to the 2016 earthquake. Despite this, the upper cliff face remained susceptible to subaerial triggers and
failed during the 2016 Kaikōura earthquake. While additional site-specific investigation into geologic
controls on cliff retreat at Conway Flat would be required to make more robust claims, lithology and
cliff geometry likely play a role in the spatial variability of retreat at Conway Flat.

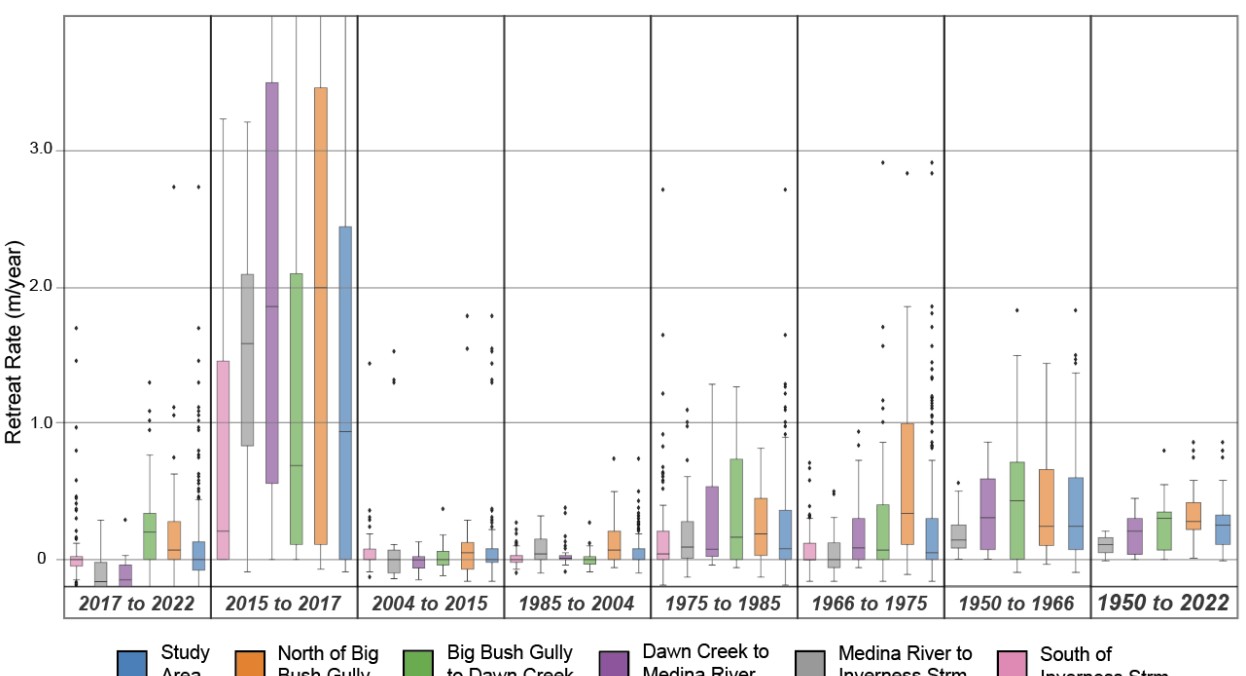

**Figure B1. Boxplots showing the spread of retreat rates within each section of coastline at Conway Flat.**

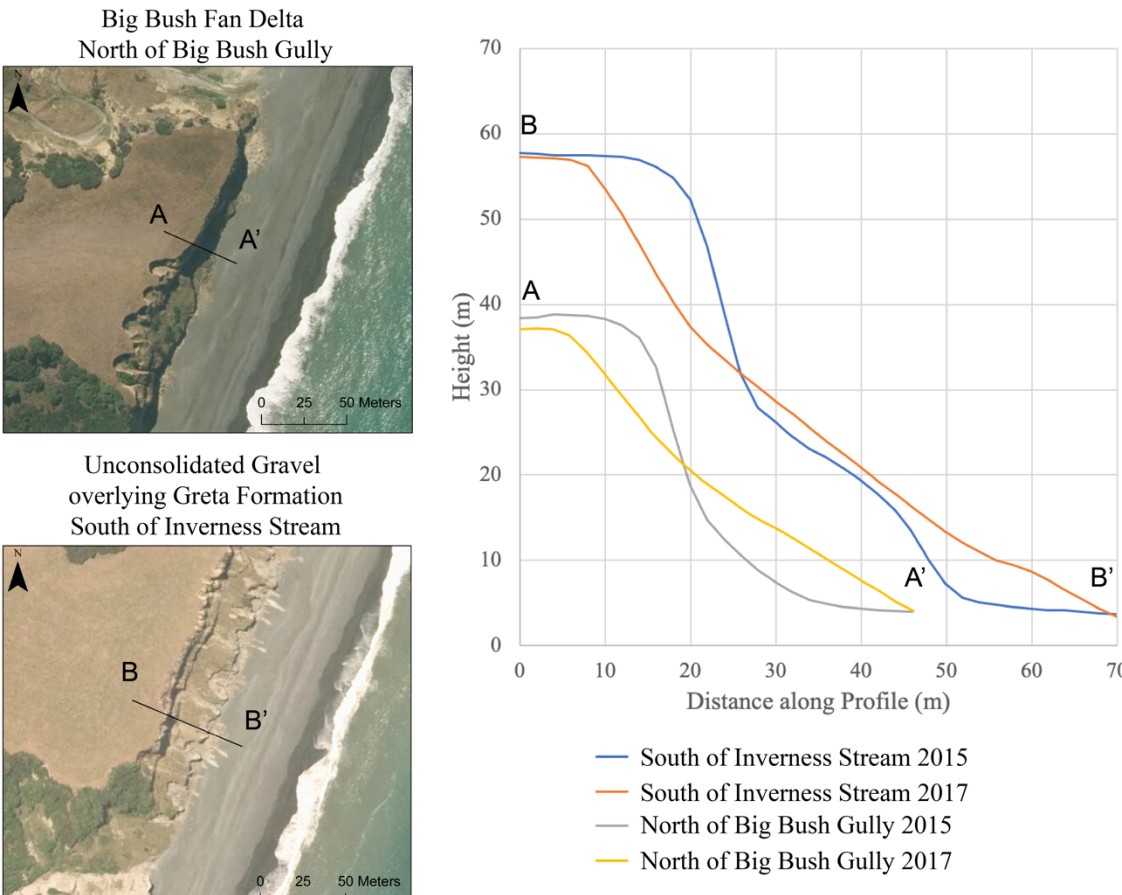

**Figure B2. Comparative Cliff Profiles. Profile A is taken from a cliff face that consists entirely of fan delta deposits. Profile B is taken from a cliff face that consists of Greta Formation overlain by unconsolidated gravels. The contact between underlying Greta and overlying unconsolidated sediment occurs at approximately 30 m height in Profile B. The location of profiles is noted on the 2015 orthoimages to the left (LINZ, 2022).**

## Appendix C – Supporting Oblique Photos

In addition to aerial images collected from the Crown Aerial Film archive (LINZ, 2021), we acquired several oblique photos of the Conway Flat cliffs prior to the 2016 Kaikōura earthquake from unpublished investigations (Figures C1-2) and collected photos of the coast following the earthquake in 2022 (Figure C3). In general, the photos support our interpretation of the pre- and post-Kaikōura earthquake cliff morphology and provide a useful visual reference. The earliest oblique photos, presented in Figure C2, were taken to document remnant native forests along the North Canterbury coast by the former New Zealand Department of Scientific and Industrial Research (DSIR). High resolution prints of the images were provided by Miles Giller of QEII Trust. Despite several attempts, we were unable to locate the photo negatives or the exact date of image collection. Based on comparison with other aerial imagery the photos appear to have been taken between 1966 and 1985.

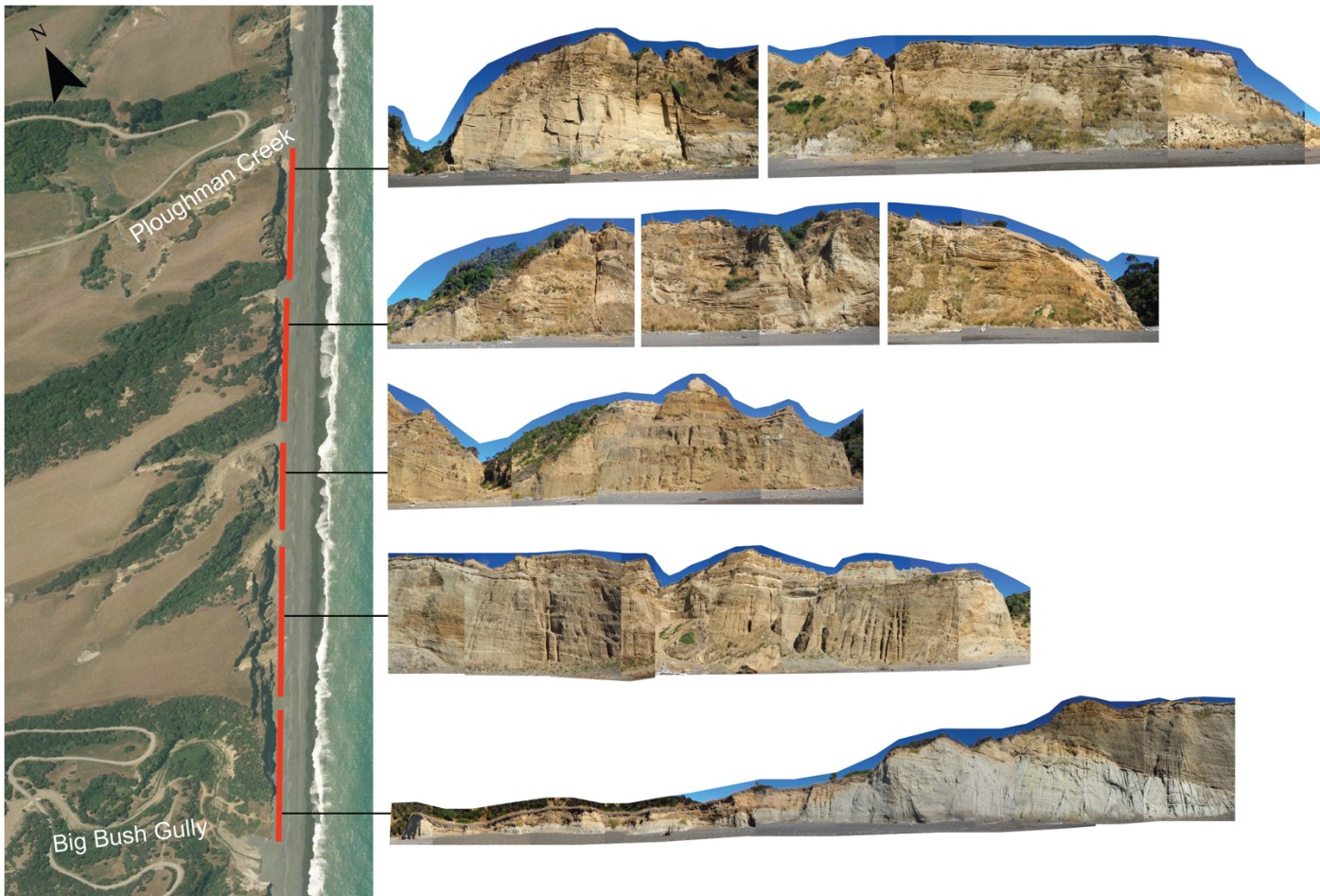

**Figure C1. Oblique photos manually stitched into mosaics of the cliffs at Conway Flat between Big Bush Gully and Ploughman Creek. Photos taken on February 17th, 2015 by Kari Bassett. A 2015 orthomosaic is provided for reference on the left (LINZ, 2022).**

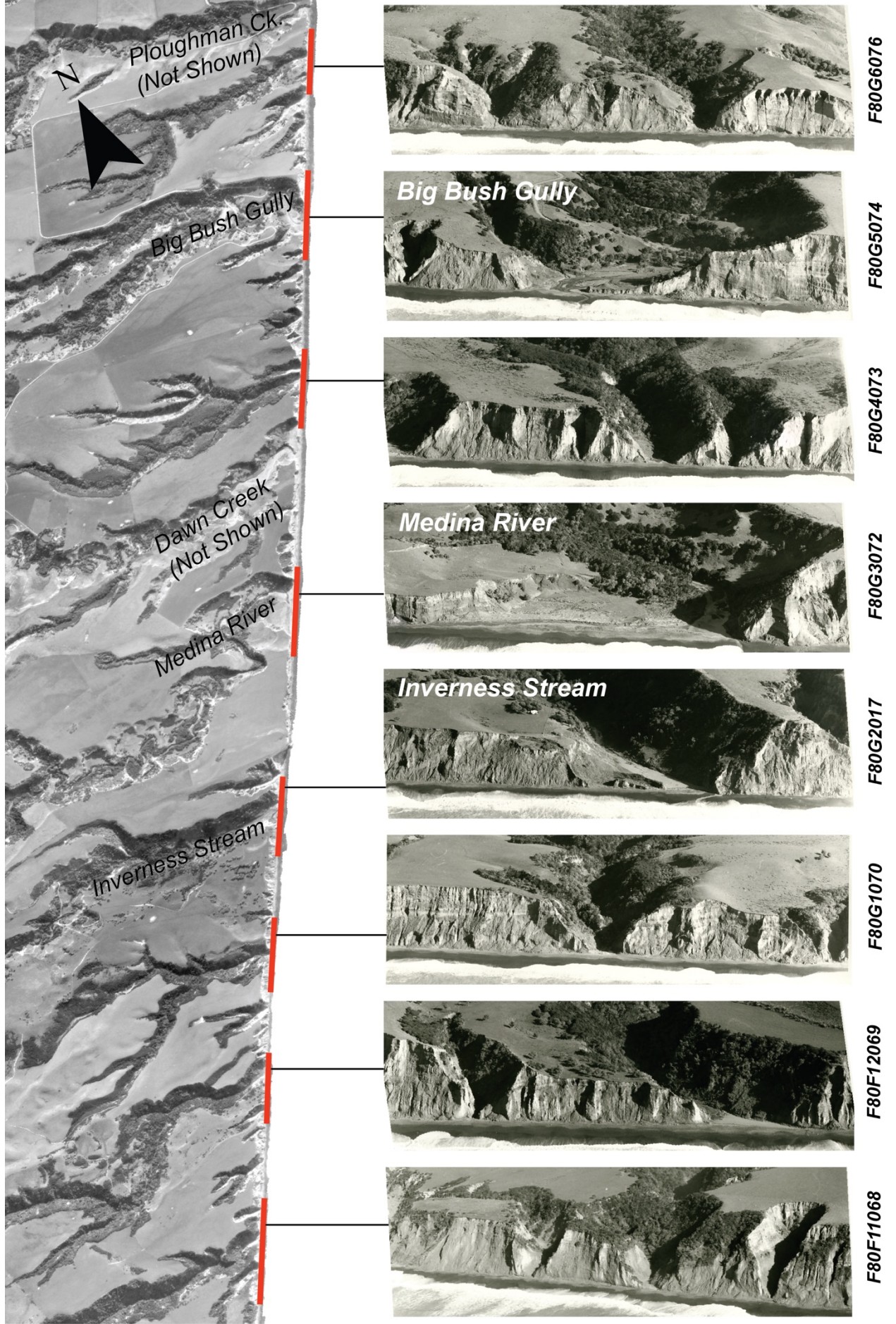

**Figure C2. Oblique photos of the cliffs at Conway Flat. Date unknown but c. 1970's based on comparison with dated aerial image sets. Photos provided by Miles Giller (QEII Trust). A 1966 orthomosaic is provided for reference on the left (LINZ, 2021).**


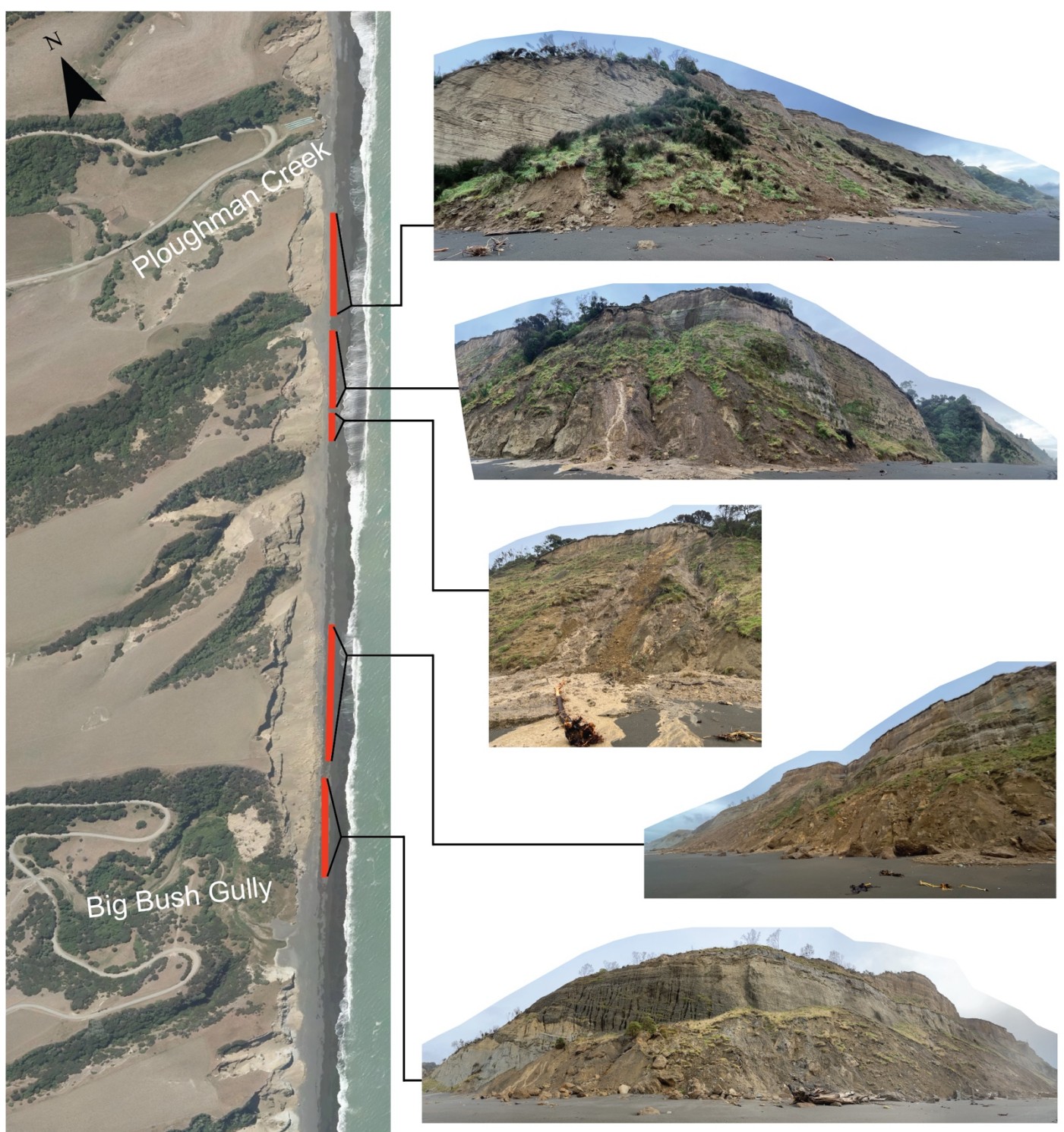

**Figure C3. Oblique photos of the cliffs at Conway Flat between Big Bush Gully and Ploughman Creek showing the state of post 2016 Kaikōura earthquake debris piles on July 28th, 2022. A 2017 orthomosaic is provided for reference on the left (Massey et al., 2020).**

**Acknowledgements**

This work was primarily funded by the New Zealand government as part of the Ministry of Business, Innovation and Employment Endeavour-funded "Earthquake Induced Landslide Dynamics" project with further support from Toka Tū Ake, the New Zealand Earthquake Commission and QuakeCoRE: The New Zealand Centre for Earthquake Resilience. The authors would like to extend a large thank you

to the landowners and residents of Conway Flat for their assistance and support in this research.

Additional thanks to Kari Bassett at the University of Canterbury and Miles Giller at QEII Trust for sharing their oblique photo datasets from Conway Flat. Thanks to Jamie Gurney who provided a summary of news articles related to the two historical earthquakes near Cheviot. Finally, thanks to reviewers Mark Dickson and Colin Murray-Wallace for their constructive comments which greatly improved the manuscript.

## Author Contributions

CB: Conceptualization, Methodology, Formal analysis, Investigation, Visualization, Writing – original draft. CS: Methodology, Investigation, Visualization, Writing – review & editing. TS: Supervision, Funding acquisition, Conceptualization, Methodology, Writing – review & editing. AH: Supervision, Methodology, Writing – review & editing. CM: Supervision, Funding acquisition, Writing – review & editing.

## Code availability

N/A

## Data availability

A .csv file with raw cliff-top retreat data from Conway Flat is included as a supplement to this manuscript.

## Competing Interests

The authors declare that they have no conflict of interest.

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
