# Peer review of "Earthquake Contributions to Coastal Cliff Retreat"

_EGUsphere, 2022_

## Author Response (AR1)

*Our responses to the two reviewer comments are included below. These are in line with our posted comments but have been updated slightly to reflect the final revised text.*

*Response to Reviewer #1:*

*Dear Professor Dickson,*

*Thank you for your thoughtful and detailed comments on our manuscript as well as the citations you provided. You raise several interesting questions that are very much worth considering.*

*In regard to your overall comment about feedbacks between cliff failure and coastal erosion, we agree that this deserves a bit more attention and we intend to provide some additional detail in our discussion. Perhaps more importantly, we believe that most of your comments can be resolved by clarifying our use of terminology. In particular, as it pertains to the terms 'long-term retreat' and 'background retreat'. In most cases, we intend to modify 'long-term' to read 'multidecadal' as we believe that this better captures our meaning and the timescales over which forecast models are focused. Additionally, we intend to modify 'background retreat' to read 'non-seismic retreat' as this term focuses on process rather than timescale. We use the term to differentiate retreat caused by rainfall, undercutting of the cliff face, and/or weathering and gravity from retreat caused by strong ground motion. The overall contribution of this paper is to demonstrate the potential for earthquakes to introduce a high degree of variability into estimates of coastal cliff-top retreat using the historical record and we maintain that our results and discussion demonstrate this well when considering multidecadal timescales.*

*Below, we engage with your comments in more detail and introduce some additional changes we propose to improve the manuscript. Each of your comments is included (or summarized) and our response follows in italics.*

I wonder how it is possible to be confident that, had the earthquakes not occurred, that other mechanisms (e.g. rainfall) would not have stimulated failure (if the cliff were over-steepened by coastal erosion)? Eventually a cliff will become steep enough that it will fail, and while this may sound provocative, does it really matter, in respect to the long-term erosion rate, what the triggering event is? In other words, over a given time scale, is the fundamental pace of cliff retreat actually governed by the coastal erosion rate (i.e. the speed at which debris is removed and at which basal steepening occurs)?

*The magnitude and variability of retreat at the cliff-toe was not systematically estimated in our analysis but, based on visual inspection of the imagery, we infer that the cliff-toe at Conway Flat is retreating more steadily than the cliff-top over the historical record. This leads us to believe that you are likely correct about the limited contribution of earthquakes (and other cliff-top triggers) over increasingly long timescales. To help us conceptualize your comments, we developed a hypothetical 300-year annual record of retreat for the top and toe of the Conway Flat cliffs (Figure R1 – See comment supplement) following the logic that coastal erosion from wave action defines cliff position over this longer timescale. This was simply a thought experiment – we were interested in how cliff-toe and cliff-top retreat rates could vary depending on the time interval being considered. The magnitudes of annual retreat (y-axis) in Figure R1 are also schematic but are loosely based on our observations from the historical record where large rainfall events (and other non-seismic triggers) cause infrequent cliff-top retreat of c. 1 to 2 m, earthquakes result in c. 4 m of cliff-top retreat c.*

*every 50 years, and it takes about 10 years after a large event to clear enough debris from the beach for wave action to resume erosion of the cliff-toe.*

[Figure]

*Figure R1. Hypothetical cliff-toe and cliff-top retreat and cumulative retreat at Conway Flat. Over multidecadal timescales, variability in cliff-top and cliff-toe retreat can result in substantially different retreat rates.*

*Over long timescales (longer than the historical and multidecadal records – for example from 0 to 300 years in Figure R1), the position of the cliff-face is dictated by coastal erosion from wave action which primarily influences the toe of the cliff. The retreat rates of the cliff-top and cliff-toe are equal. Over this timescale, earthquakes and other subaerial triggers unrelated to wave action indeed have little influence on the overall position of the cliff. Over decadal and multidecadal timescales, i.e. those considered in the historical record and forecast models (for example 0 to 50 or 50 to 100 years in Figure R1), the rate of cliff-top retreat can vary substantially from retreat at the toe of the cliff.*

*Thus over these timescales, capturing the contribution of e.g. earthquake triggering mechanisms in models becomes quite important for characterising cliff-top retreat rates.*

I wonder whether the paper is weighing up the relative importance of different failure event drivers, rather than earthquakes versus coastal erosion? I think this nuance is quite important. For instance, what do you think would be the effect of several large earthquakes in quick succession? Presumably the cliffs would fail toward a stable angle and further erosion would not be unlikely without basal debris removal and oversteepening? In this case, can you be sure that the long-term cliff retreat rate would be increased?

*We also agree that some removal of debris and oversteepening of the cliff face at Conway Flat is required to make the cliff-top susceptible to failure. In theory, rapidly successive earthquakes and rainfall events could outpace coastal retreat from wave action and the cliff-top could become less steep and less susceptible to failure. This process would slow cliff-top retreat and give coastal erosion from wave action at the toe the chance to 'catch up'. This may explain the less steep cliff geometry and lower historical retreat observed south of Inverness Stream (Figure B1). However, in many cases, for example in the fan delta material north of Big Bush Gully, the time required to return the cliff-top to a susceptible geometry between large subaerial triggering events appears to be much lower than the 50-year average return interval for strong ground motion. Between 1950 and 1966 we observed c. 5 m of average retreat at Conway Flat with little evidence of this retreat present in 1966. Within c. 15 years, the cliffs at Conway Flat were once again steep and likely susceptible to further widespread failure.*

*It is worth mentioning that the degree to which oversteepening is required for failure is dependent on the material properties of the cliff face, the means of triggering a failure, and the failure mechanism. For example, dynamic stresses introduced during an earthquake may result in failure at a less susceptible geometry than would be required for non-seismic failure. On average, 2016 coseismic failures along the Kaikōura coast occurred on c. 40° slopes, much lower than the near vertical slopes present at Conway Flat (Bloom et al. in Review). Additionally, in some cases, failure mechanisms at Conway Flat resulted in steep headscarps that do not preclude subsequent failure at the same location over decadal timescales. Figure 6 in the manuscript shows images of a portion of the cliff face before and after the 2016 Kaikōura earthquake. The debris avalanche at this site has resulted in a steep section of cliff-top that is still highly susceptible to strong ground motion or rainfall induced failure following the 2016 earthquake.*

*To address your first two comments in the manuscript, we intend to add text at the end of the first paragraph in the introduction (revised lines 39 to 41) reading: "This is particularly important when considering that the cliff face may erode at different relative rates over decadal to multidecadal timescales, for example subaerial influences eroding the cliff-top faster than coastal erosion from wave action at the toe of the cliff." Additionally, we intend to add a paragraph at the beginning of section 5.3 (revised lines 389 to 395) which reads: "Over long timescales (i.e. longer than the historical record), the rate of coastal erosion at the base of the Conway Flat cliffs may limit the extent of cliff-top retreat and, in turn, the long-term influence of subaerial triggers like earthquakes or rainfall. This is because some oversteepening of the cliff face is likely a prerequisite for cliff-top failure (Wolters and Müller 2008). Prior to 2016, most of the cliff face at Conway Flat was near vertical in many places (Figure 5), an indication of dominant marine erosion (Emery and Kuhn 1982). However, over multidecadal timescales, the rate of cliff-top and base retreat may vary substantially due to the greater temporal variability of subaerial triggers."*

Useful attempts to extend the work are provided in Equation 1 and the model illustrated in Fig 8. Is equation 1 assumed to be independent of the pre-existing shape of the cliff? If the cliff was already over-steepened, then the magnitude of shaking required to cause failure would be lower, correct? I found the model in Fig 8 interesting and I can see how the combination of high coastal erosion and low earthquake frequency means that earthquakes have limited influence on the retreat rate, but I'm not clear on how low coastal erosion and high earthquake retreat combines in the way you allude to, because I guess I am unconvinced that feedbacks have been accounted for (e.g. high sed supply to the coast and failure toward a more stable slope).

*Equation 1 is a first-order estimate and is unlikely to be very useful in cases where the cliff-top is not susceptible to earthquake induced failures (for example if the cliff-top is not very steep). We intend to revise the text on line 412 (revised line 456) to read: "Using these three inputs, and assuming a cliff-top susceptible to failure, a simple equation can be defined …". Assuming there is a susceptible cliff-top available for failure, low coastal erosion and high earthquake retreat could result in earthquakes having a substantial influence on multidecadal cliff-top retreat. When erosion rate is integrated over longer time frames, we agree that earthquakes (and other subaerial triggers) are likely subordinate to preconditioning via wave erosion at the toe of the Conway Flat cliffs.*

The introduction section is quite short. I think the potential implications of the research presented would be enhanced if the authors were to broaden the framing a little (e.g. some possible papers to consider below). However, the essential point re tectonic contributions to cliff retreat is well made in the introduction.

*In looking through the citations that you include, we agree that there is some room to bolster the framing of our introduction with additional references. We intend to revise the first paragraph of the manuscript to read: "As sea level rises from climate change, regional coastal modeling suggests an increasing rate of coastal cliff retreat (e.g. FitzGerald et al., 2008; Limber et al., 2018). This increasing retreat rate will pose a significant hazard to people and property around the globe, particularly in regions that face a high risk due to population exposure (e.g. He and Beighley 2008). The response of individual coastal cliffs to sea level rise is complicated by a range of feedbacks and site specific conditions, for example changing beach volume, the transport of failed material from more erosive sections of coastline, and cliff material strength (e.g., Dickson et al., 2007; Ashton et al., 2011) as well as by the temporal variability of cliff retreat (e.g., Hall et al., 2002; Hapke and Plant 2010). Many decadal to multidecadal models of coastal cliff retreat rely heavily on historical records and legacy aerial imagery (typically less than 50-100 years) for calibration, in part to capture some of this spatial and temporal variability (e.g. Dickson et al., 2007; Young et al., 2014; Limber et al., 2018). Unfortunately, direct evidence of past coastal failures is rarely preserved in the active coastal environment (Francioni et al., 2018) making it difficult to confirm that the historical record is representative of all possible preconditioning and triggering mechanisms for coastal cliff collapse. This is particularly important when considering that the cliff face may erode at different relative rates over decadal to multidecadal timescales, for example subaerial influences eroding the cliff-top faster than coastal erosion from wave action at the base of the cliff."*

70 - label Conway Flat on Fig 1

*This will be added in the revised manuscript.*

 box label the position of Fig 2 on Fig 1

*The blue outline currently included and labeled in Figure 1 is the position of Figure 2.*

108 - wording "have had limited to no anthropogenic modification"

*Commas will be added here (revised line 149) to read: "...have had limited, to no, anthropogenic modification…"*

The background section is lacking detail on uplift (if any) within the study site? This is important in relation to the level of wave action in relation to the cliff toe. Can some detail be added in this regard.

*Revised Lines 138 to 145 now further discuss coastal uplift in the Kaikōura region: "Ota et al. (1996) suggested c. 2 to 3 mm/year of Holocene uplift along the Conway coast based on marine terrace heights and radiocarbon ages collected from buried trees within the Ngaroma Terrace north of Ploughman Creek (Figure 2). Recent recalibration of these radiocarbon dates by Barrell et al. (2022) suggest that regional tectonic uplift is closer to 1.2 mm/year. While these rates of tectonic uplift are loosely constrained, they generally agree with estimates of tectonic uplift (c. 0.9 to 1.2 mm/year) in marine terraces further north on the Kaikōura peninsula (Nicol et al., 2022). Site-specific uplift rate estimates are currently poorly constrained south of the Ngaroma terrace and Ploughman Creek."*

*In order to make the importance of uplift clear we will also add the following sentence to the discussion in section 5.1 (revised lines 360 to 364): "In addition to lithology, several other factors may further influence the rate of local cliff retreat at Conway Flat over the historical record but these are more challenging to quantify. For example, the rate of tectonic uplift and sediment transport may influence beach height in relation to the base of the cliffs at Conway Flat (e.g., Horton et al., 2022) but it is not possible to quantify these changes across our historical image datasets."*

3.1 and Appendix A - can you provide a little more detail in regard to the appropriateness of using the 2017 DSM for historic photographs? As you demonstrate in the paper, there is significant erosion and landsliding between epochs, but it's not possible to have a DSM for each time period in the historic photograph record - to what extent is this accounted for in the uncertainty estimation?

*We reference all images horizontally to 2017 aerial imagery and vertically to a DEM generated from the 2017 aerial imagery. While we could have used any of the aerial images as a base for our referencing, the 2017 aerial imagery was chosen based the quality and extent of the dataset (this will be noted in Appendix A). As currently discussed in Appendix A starting at line 505, georeferencing uncertainty was considered based on the residuals between control points in each image and the 2017 imagery. Because each image is referenced to the 2017 image we can include a consistent estimate of georeferencing uncertainty at these control points (typically fence lines, building corners, and stock ponds) which were assumed to be relatively unchanged between image epochs. As you point out, erosion and landsliding between epochs makes it difficult to estimate uncertainty directly along the coastline. Instead we interpolate between our unchanged control points to generate a unique estimate of georeferencing uncertainty for each measurement along the cliff-top.*

175 - does the presence of dense vegetation preclude the possibility of cliff retreat - in other words, how were you confident there was no cliff retreat if you couldn't see the cliff because of dense vegetation? "where dense vegetation was present across all epochs of imagery and we were confident that no cliff retreat had occurred"

*No, the presence of dense vegetation across all epochs does not completely preclude the possibility of cliff retreat but it does suggest that cliff retreat was limited. In most cases where we observed a loss of dense native vegetation due to failure of the cliff-top, some evidence of this failure remained present across the following image epochs. We believe that it is unlikely that significant cliff-top failure could occur and be completely re-obscured over the limited time between our imagery. We intend to revise line 175 (revised line 187) to read "...where dense vegetation was present across all epochs of imagery and we were confident that no significant cliff retreat had occurred..."*

245 - can you be a little clearer with your wording regarding failure/no failure of the Greta Formation? Can you provide some numeric stats on how much of this section failed? This would be useful as a lot of the literature on soft-rock coasts deals with consolidated fine-grained rock as distinct to the (unconsolidated?) overlying delta deposits.

*Most failures from the 2016 earthquake occur within overlying unconsolidated fan delta deposits. We only observe a few areas where there is clear evidence of failure within the Greta Formation. Quantifying this regionally is quite challenging and, while we agree that it is worthwhile thinking about the relative contributions of the Greta Formation to cliff retreat at Conway Flat, we believe providing these statistics is largely beyond the scope of the current manuscript. We intend to add the following text on line 249 (revised line 261): "Further site-site specific investigation, largely beyond the scope of this work, would be required to further elucidate the relative contribution of the Greta Formation to failures at Conway Flat."*

260 - the Medina fan deposit is 'more consolidated' - more than the Dawn fan delta deposits? Is there a way to characterise the degree of consolidation of these different deposits? It would be useful to put these erosion rates into context of other cliff erosion rates reported iternationally. You state that these more consolidated cliffs have eroded less - do you ascribe this to their degree of consolidation?

*To first order, yes, we infer that the consolidation of the Median fan deposit may have resulted in less erosion than the Dawn fan delta deposit. As in the comment above, additional geotechnical investigation would be required to quantitatively define the consolidation or void ratio of the various fan delta deposits at Conway Flat and directly relate erosion rate to material properties.*

325 - I find this statement confusing: "it does appear that more indurated material (with assumed higher shear strength) in the lower cliff face may buffer the upper cliff face from wave action effectively reducing the background rate of cliff-top retreat (Emery and Kuhn 1982)". Are you trying to say that the presence of more resistant prevents over-steepening of the cliff face, and that this then manifests in slower cliff-top erosion rates? Can you provide some cliff-profile sections? These will presumably show that where the mudstone occurs the lower cliff is steeper (because of harder rock), in comparison to the sections where the fan deposits extend to the cliff toe. I think these profiles would help support the argument you are developing here.

*We are inferring that, in some sections of the Conway Flat coast, lower historical cliff-top retreat could be a result of more resistant geology at the toe of the cliff that prevents undercutting of the upper cliff by wave action (we will replace 'background retreat' with 'non-seismic retreat'). Over multidecadal timescales this may result in subaerial triggers like earthquakes playing a more important role in cliff-top retreat than wave action. As described in a previous comment, when cliff-top retreat outpaces retreat at the toe of the cliff, the steepness of the cliff-top can reduce, effectively slowing cliff-top retreat but this is not always the case at Conway Flat. Figure R2 (see response*

*supplement) provides comparative profiles demonstrating this. In profile A to A' in Figure R2 the cliff-top is influenced by both wave action at the toe of the slope (pre-2016) and subaerial triggers at the top of the slope. Alternatively in profile B to B', terracing occurs with Greta formation buffering cliff-top retreat from undercutting by wave action prior to the 2016 earthquake. Despite this, the upper cliff face remained susceptible to subaerial triggers and failed during the 2016 Kaikōura earthquake.*

[Figure]

*Figure R2. Comparative Cliff Profiles. Profile A is taken from a cliff face that consists entirely of fan delta deposits. Profile B is taken from a cliff face that consists of Greta Formation overlain by unconsolidated gravels. The contact between underlying Greta and overlying unconsolidated sediment occurs at approximately 30 m height in Profile B.*

*We believe that additional site specific investigation into geologic controls on cliff retreat at Conway Flat would be required to make more robust claims but we are happy to include Figure R2 alongside Figure B1 in the appendix of the revised manuscript.*

5.2 - the section on sediment loss from the cliff-toe alludes to storms, and these may well be important, but variability in longshore sediment transport processes could drive much of the natural variability in beach volume - there is an extensive literature related to this. Is there any evidence to

suggest that the 1960 tsunami runup resulted in significant sediment transport on beaches in NZ - not that I am aware of? On line 381 you more directly suggest that tsunami debris removal has occurred, but I don't think there is evidence of this?

*This is a valid point and we will modify the text on lines 344 to 345 (revised lines 377 to 379) to read "The extent to which storm surge, from events like Ex-tropical cyclone Gita (Figure 7), and variability in longshore sediment transport (Larson and Kraus, 1993; Dickson et al., 2007; Karunarathna et al., 2014) influence the removal of failed debris at Conway Flat remains largely unclear…" Additionally, we do not have direct evidence of sediment transport from the 1960 tsunami runup at Conway Flat but we cannot completely rule this out. We will modify the text on line 381 (revised line 424) to read "… that may have enhanced…"*

360 - "Prior to 2016, most of the cliff face at Conway Flat was near vertical in many places (Figure 5), an indication of a combination of subaerial and marine erosion (Emery and Kuhn 1982)." Here Emery and Kuhn are implying a dominance of marine over subaerial processes.

*We will modify the text on line 381 (revised line 392) to read: "Prior to 2016, most of the cliff face at Conway Flat was near vertical in many places (Figure 5), an indication of dominant marine erosion (Emery and Kuhn 1982)."*

379 - "Assuming little change in the background rate of coastal erosion between time windows" - here I think you are referring to the removal of basal debris, rather than cliff-toe steepening? Would it be useful for you to refer to beach erosion and cliff-toe erosion to distinguish what you are referring to when you say "coastal erosion"?

*We agree that our use of terminology could introduce some confusion here. We will modify the text on line 379 (revised line 421) to read: "Assuming little change in the rate of beach erosion between time windows…"*

***Response to Reviewer #2:***

*Dear Professor Murray-Wallace,*

*Thank you for your constructive comments. We have responded to your suggestions below (our responses are in italics), and we have made some additional changes to the manuscript.*

This is a very interesting manuscript on the nuanced issue to quantifying historical records of coastal cliff retreat based on an example from the Conway Flat area on the South Island of New Zealand. The manuscript is well-written and addresses many of the conceptual and methodological issues of quantifying coastal cliff retreat from time-series images.

For people unfamiliar with the area, it would be nice to have a few photographs illustrating the general nature of the coastal cliffs. This would help in part to understand the physical nature of the cliffs and for the reader to infer potential coastal processes responsible for the changing form of the cliffs through time. This would be good particularly for people unfamiliar with this coastal sector.

*We agree that additional photos may add value for some readers and have included a selection of photos in Appendix C. Figure 6 in the manuscript includes before and after photos from the 2016 Kaikōura earthquake that should allow more expeditious readers to evaluate the nature of the Conway Flat cliffs.*

The manuscript could potentially be strengthened by some description and discussion of the inherent structural integrity of the lithological units on which the cliffs have developed. Apart from the deltaic succesions, can some commentary be made about the other bedrock units in terms of the broad, regional-scale structural characteristics such as fault , joint or cleavage density and trends? Some discussion on whether some lithologies breakup in a predictable manner or in a more random fashion? - Unilinear or non-linear response to strain irrespective to earthquakes?

*Some additional information on the underlying Greta formation bedrock will be added to the manuscript in section 2.2 (revised lines 86 to 88): "At Conway Flat, situated between the Conway and Waiau river mouths (Figure 2), weak Neogene Greta Formation mudstone (Uniaxial Compressive Strength <2 MPa) with massive near horizontal bedding (Rattenbury et al., 2006; Massey et al., 2018) is overlain by…" and section 5.1 (revised lines 339 to 345) as follows: "In the Kaikōura region and across New Zealand, failures in tertiary sediment including the Greta Formation mudstone tend to occur as large planar slides often failing along preferentially oriented bedding planes (Pettinga, 1987; Mountjoy and Pettinga, 2006; Singeisen et al., 2022) or as shallow debris avalanches in more weathered sections of the rock mass (Massey et al., 2018). We do not observe evidence of planar sliding at Conway Flat over the historical record and most retreat of the underlying Greta formation appears to result from shallow debris avalanching, observed in some aerial imagery, alongside more gradual erosion due to wave action."*

Perhaps modify Section header 5.1 as 'Geology' is a discipline rather than a descriptor of bedrock characteristics or lithologies.

*We will modify the Section 5.1 header to read: "Cliff Retreat and Lithology"*

I wondered if some commentary can be made where the sediment ends up post cliff collapse? It may be useful to have some commentary on this matter. Does the sediment end up on the continental shelf below storm weather wave base, or is it in part, transported along shore? If the latter, does beach nourishment protect the backing cliffs in some localities.

*The nature of sediment transport at Conway Flat is an interesting question with no clear answer given our current analysis. Targeted field and remote sensing investigation, largely beyond the scope of this work, would likely be required to robustly evaluate sediment transport along the Conway coast. This being said, we do believe that this is an important topic to acknowledge, and we will add the following text in Sections 1 (revised lines 29 to 33) and 5.2 (revised lines 377 to 380) respectively:*

*"The response of individual coastal cliffs to sea level rise is complicated by a range of feedbacks and site specific conditions, for example changing beach volume, the transport of failed material from more erosive sections of coastline, and cliff material strength (e.g., Dickson et al., 2007; Ashton et al., 2011) as well as by the temporal variability of cliff retreat (e.g., Hall et al., 2002; Hapke and Plant 2010)."*

*"The extent to which storm surge from events like Ex-tropical cyclone Gita (Figure 7) and variability in longshore sediment transport (Larson and Kraus, 1993; Dickson et al., 2007; Karunarathna et al., 2014) influence the removal of failed debris at Conway Flat remains largely unclear due to our limited number of image epochs…".*

I also wondered if it is appropriate to have some commentary on the general aspect of the coastline at a more detailed level to consider contrasting wave attack and erosion? Are some sectors of the coastline more prone to erosion, therefore increasing the likelihood of coastal retreat irrespective of the influence of earthquakes?

*Aspect of the coastal cliffs at Conway Flat likely has some influence on wave attack and erosion. In concert with variability in lithology, this may help to explain some local variability in retreat rate but does little to explain widespread retreat observed following earthquake events. We will add the following text in Section 5.1 (revised lines 364 to 366) as follows: "Likewise, local aspect of the cliff face in relation to variable incoming wave direction may influence the rate of cliff retreat but information on changes in wave direction through time are unavailable."*

Line 393 - can some supporting information be provided about the validity of the estimate of long-term cliff-top retreat?

*As discussed in more detail in our response to Reviewer 1, we will change 'long-term' to 'multidecadal' throughout the manuscript as this better represents the timescales over which cliff-top retreat may vary from overall cliff position. Given the average return interval of shaking at Conway Flat (c. 50 years), including retreat from two earthquakes is unlikely to be representative of true multidecadal retreat.*

Line 406 '... compared with retreat ...'

*We will revise this statement accordingly.*

Indicate place name 'Conway Flat' on  Figure 1.

*A label will be added.*

---

## Author Response (AR3)

In addition to changes made to address reviewer comments in our previous manuscript iteration, we have made some minor changes in this iteration to address typos and inconsistencies in figure numbering.

---

## Author Response (AR4)

The coauthors have checked the manuscript and are happy for it to proceed to copy editing.